# Event segmentation applications in large language model enabled automated recall assessments
Ryan A. Panela [1,2] ✉, Alexander J. Barnett [2,3], Morgan D. Barense[1,2] & Björn Herrmann [1,2] ✉

Understanding how individuals perceive and recall information in their natural environments is critical to understanding potential failures in perception (e.g., sensory loss) and memory (e.g., dementia). Event segmentation, the process of identifying distinct events within dynamic environments, is central to how we perceive, encode, and recall experiences. This cognitive process not only influences moment-to-moment comprehension but also shapes event specific memory. Despite the importance of event segmentation and event memory, current research methodologies rely heavily on human judgements for assessing segmentation patterns and recall ability, which are subjective and time-consuming. A few approaches have been introduced to automate event segmentation and recall scoring, but validity with human responses and ease of implementation require further advancements. To address these concerns, we leverage Large Language Models (LLMs) to automate event segmentation and assess recall of written narratives, employing chat completion and text-embedding models, respectively. We validated these models against human annotations and determined that LLMs can accurately identify event boundaries, and that human event segmentation is more consistent with LLMs than among humans themselves. Using this framework, we advanced an automated approach for recall assessments which revealed semantic similarity between segmented narrative events and participant recall can estimate recall performance. Our findings demonstrate that LLMs can effectively simulate human segmentation patterns and provide recall evaluations that are a scalable alternative to manual scoring. This research opens avenues for studying the intersection between perception, memory, and cognitive impairment using methodologies driven by artificial intelligence.

Research in many domains, ranging from perception and memory, is concerned with how humans process information in everyday environments[1–4]. Although environments unfold continuously over time, one dominant framework, the *event segmentation theory*, suggests that individuals discretize or segment experiences into meaningful *events*[2,3,5]. Events can span vast time-scales—from a few seconds in a short conversation to baking cookies over an hour[2,3,6]. Event segmentation is thought to be fundamental to how we perceive and mentally structure our experiences for effective future recall[2,3,5,7–9], and can influence failures in perception and memory[10,11]. In this paper, we focus specifically on written narratives and spoken recall transcripts to extend and validate methods for investigating event segmentation and subsequently leverage its properties for application in recall assessments.

Historically, event segmentation has been studied by having participants read text or watch a movie and simultaneously identify the *boundaries* between events with markings or button presses[12,13]. Event segmentation is subjective[8], but it has been demonstrated that participants tend to mark event boundaries at similar locations, known as *normative boundaries*[10,13,14]. This across-participant agreement supports the growing practice of using segmentation data from one group to examine the cognitive performance in another[9,14]. Boundaries from individuals that closely align with the normative boundaries, as evidenced by high group agreement, demonstrate sensitivity to meaningful event features, suggesting that they may exhibit stronger cognitive performance and subsequent memory of those events[8,9,15]. Since event segmentation and memory are functionally

[1]Rotman Research Institute, Baycrest Academy for Research and Education, North York, Toronto, ON, Canada. [2]Department of Psychology, University of Toronto, Toronto, ON, Canada. [3]Department of Neurology and Neurosurgery, Montreal Neurological Institute and Hospital, McGill University, Montreal, QC, Canada. ✉e-mail: ryan.panela@utoronto.ca; bherrmann@research.baycrest.org

integrated, studying these processes together may not only enhance our understanding of how experiences are structured and retained but also provide a framework for assessing memory processes more broadly.

Given the importance of event segmentation in memory research, practical considerations arise when implementing segmentation-based methods in experimental design. Research questions and applications may differ depending on whether understanding human event segmentation is the primary focus or whether a researcher aims to identify event boundaries for analytical purposes (e.g., recall) or stimulus manipulation (e.g., standardizing the number of events across stimuli). Critically, manually segmenting experimental stimuli into discrete events can be time-consuming and financially costly, and segmentation can vary across individuals due to differing interpretations of instructions, ambiguous tasks, lapses in attention, or erroneous button presses[8,14,15]. This can be a prominent challenge if segmentation is performed by a few individuals, leading some researchers to recruit a large number of participants through online platforms[14,16], which again can be expensive. Automated event segmentation could possibly mitigate such costs and reduce data-acquisition time, especially if the purpose of identifying event boundaries is for analysis or stimulus development.

Recent advancements in Large Language Models[17–20] offer a promising avenue for automating event segmentation. Cognitive neuroscience has begun to leverage LLMs by investigating their alignment to human behaviour[17,21–24] and brain activity[25–28]. Recent work suggests that OpenAI's GPT-3 model[29] can be used to automate event segmentation resembling that of human participants[1]; however, OpenAI's GPT is proprietary and associated with fees paid through an online account whenever input is fed into the model through the Application Programming Interface (API). Other powerful models, such as Meta's LLaMA 3.0[20], are free and can be downloaded and used offline. LLaMA could possibly enable cost-sensitive, privacy-forward, offline applications. This makes LLaMA particularly useful when working with datasets that involve identifiable human data (e.g., free recall, see below) or large-scale text analyses, where avoiding API costs is beneficial. LLaMA 3.0 has been shown to be comparable to OpenAI's GPT model on some benchmarks[30–32], but it is unclear whether LLaMA can be used for event segmentation purposes. Moreover, although some analyses have been conducted in the previous work to compare GPT-3 performance to human event segmentation[1], more detailed analyses and comparisons involving critical model parameters that determine randomness of model outputs, newer LLM models (i.e., GPT-4), and links to human perception are needed to further validate the effectiveness of automating event segmentation.

Beyond perception, event segmentation plays a fundamental role in structuring memory[5,8–10], raising the question of whether automated segmentation can capture these memory-relevant structures. Research suggests that individuals whose segmentation aligns more with a group tend to have better recall performance[8,9,15], reinforcing the idea that segmentation plays a critical role in episodic memory encoding[8,33]. Episodic memory is concerned with the organization of spatiotemporal information, which is inherently structured through the segmentation of events[33–35]. If LLMs segment events in a way that resembles human perception, this suggests they capture meaningful units of experience that structure memory. By leveraging LLMs to automate event segmentation, stimulus materials, and participants' recall data can be used to examine the relationship between segmentation and recall in a more efficient and accurate manner.

Given the role of event segmentation in memory, evaluating recall often depends on structured event units; however, previous methods for assessing recall have relied on the manual segmentation of stimuli (e.g., text, audiovisual) into events[16,36–38], which can present several challenges. Human raters manually judge the accuracy of each recalled event through gist scoring[39,40], detail counts[38,41], or point-biserial coding (binary scoring; recalled or not recalled)[16,37]. Regardless of the specific approach, manual scoring is time-consuming and financially costly, which hinders scalability. Manual scoring guidelines and approaches may differ between research groups, possibly leading to inconsistencies in the literature and reproducibility challenges. A few recent approaches have been developed to automate

recall scoring[42,43]. One comprehensive approach relies on both topic modeling to obtain embeddings (numerical vectors) for text pieces and hidden Markov models (HMMs) to segment speech into events[42,44]. Several recall metrics have been developed to show the sensitivity of the approach for recall scoring[42,45]. Other research has employed related methodologies, but focused on short narratives with predefined details and clauses within narratives[46–49]. These units are much shorter than events and are unlikely to represent memory structures for everyday activities or conversations[50–54]. Critically, topic modeling and HMMs are potentially more complicated to implement compared to the few lines of Python code needed to obtain text embeddings and chat completions from modern LLMs. Depending on the model used, LLMs may also provide analysis of segmentation and recall with the same model, possibly streamlining analytic approaches.

In the current research, we leverage LLMs to automate event segmentation and recall assessments using narrative texts and their corresponding transcribed spoken recall. We build on previous research[1] to (i) investigate the capacity of newer LLMs to segment narratives into meaningful units, (ii) examine the effect of the critical randomness parameter (temperature), (iii) develop new analysis metrics, and (iv) provide further validation from human data. While each of these components— automated segmentation and automated recall—builds on prior work, a key contribution of the present study is their integration and expansion. We use LLM-generated event boundaries to not only assess their alignment with human segmentation but also as anchors for evaluating free recall, thereby establishing a unified framework that links perception to memory. To implement the automated segmentation approach, we use OpenAI's GPT-4[19] and Meta AI's LLaMA 3.0[20] to compare state-of-the-art proprietary and free models, respectively. Subsequently, to implement an automated recall assessment, we examine the effectiveness of various text-embedding models. Building on recent work[37], we compare OpenAI's proprietary text-embedding model[55], as well as free models like Google's Universal Sentence Encoder(USE)[56], Language-agnostic BERT Sentence Embedding (LaBSE)[57], and Masked and Permuted Pre-Trained for Language Understanding (MPNet)[58]. Ultimately, this work extends prior methods to provide an end-to-end framework for automating segmentation and memory recall assessments, enabling scalable analyses of how structured experiences influence memory.

## Methods
### Participants
Thirty-one younger adults participated in the current study across two different procedures. Twenty individuals participated in a narrative-reading and narrative-recall experiment to assess event segmentation and free recall ($M_{age}$ = 21.8, range: 18–33 years, $N_{female}$ = 17, $N_{male}$ = 3). The sample size was determined based on prior event segmentation research, which has demonstrated robust stability in segmentation patterns with similar or smaller sample sizes[14]. A separate group of eleven participants ($M_{age}$ = 19.81, range: 18–26 years, $N_{female}$ = 10, $N_{male}$ = 1) took part in a subsequent experiment to rate the degree to which specific locations in the narrative that were identified by an LLM felt like an event boundary. Details about each experimental procedure are described in subsequent sections. Across both samples, participants represented a diverse ethnocultural background, including individuals who identified as White or Caucasian (32%), Southeast Asian (26%), South or Central Asian (16%), African (16%), and mixed or other backgrounds (10%). One participant did not report ethnicity. Participants were either native English speakers or highly proficient English speakers who had learned English before the age of five. All participants reported having no known neurological diseases. Demographic data was self-reported by participants. Participants provided written consent prior to the study and received compensation of $10 per 30-min of participation or through course credit at the University of Toronto. The study was conducted in accordance with the Declaration of Helsinki (Version 2004) and the Canadian Tri-Council Policy Statement on Ethical Conduct for Research Involving Humans (TCPS2-2014), and was approved by the Research Ethics Board of the Rotman Research Institute at Baycrest

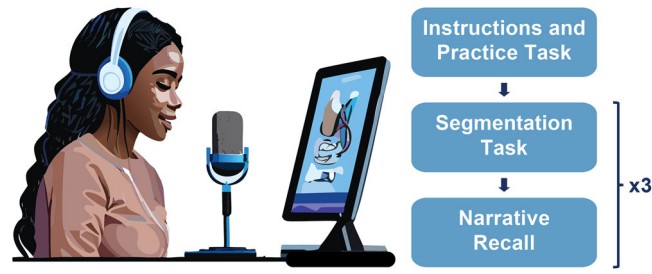

**Fig. 1 | Experimental design.** Participants initially segmented and recalled a short narrative to ensure they understood instructions. Subsequently, they read three narratives, identifying the largest event units throughout, and immediately freely recalled the narratives by speaking into the microphone.

Academy for Research and Education (REB #23-11). This study was not preregistered.

## Main Data: narrative event segmentation and recall

We used narratives extracted from Trevor Noah's memoir *Born a Crime*[59]. The book highlights his experiences growing up during the era of Apartheid in South Africa. Each chapter provides a cohesive narrative intended to create an absorbing and enjoyable reading experience for the participant. We used the following three chapters: *Run!*, *Go Hitler*, and *My Mother's Life*. Chapters were truncated such that each narrative was approximately 1500 words while maintaining narrative closure. Additionally, the chapter *Robert* was used as a practice narrative to introduce the experimental procedure (~400 words; Fig. 1). Narratives were printed on 8 ½ inch × 11 inch paper in one continuous body of text with original punctuations present, but without paragraph or other text formatting that could facilitate the identification of specific segmentation locations[60].

Instructions were presented through three modalities to ensure clarity: printed handouts, verbal guidance, and visual cues on a computer screen using PsychoPy 2023.2.3[61]. Participants were informed that the practice block would involve reading the same narrative two times and that their primary task would involve identifying events within the narrative. Specifically, an *event* was defined as a discrete piece of information that could be described as having a clear beginning and an end. To demarcate events, participants were instructed to draw a line between two words whenever they perceived one event concluding and another beginning. Importantly, participants were told that there were no objectively correct answers and that responses were subjective, allowing for individual variation. Participants received a printed copy and were asked to identify both small and large event units[5,9,15,62]. In the first reading, they focused on the smallest natural and meaningful event units. In the second reading, they shifted their attention to larger event units. This dual approach aimed to provide participants with a nuanced understanding of narrative levels[5,9,15] as the main procedure instructed participants to focus solely on the large event units.

After completing the segmentation task, participants provided a free recall of the practice narrative. They were instructed to speak into a microphone (Shure SM7B; Steinberg UR22C external sound card) and provide as much detail as possible, even if they believed some details were insignificant to the narrative progression. The free recall for the practice narrative was performed twice. Participants first recalled the narrative, and once they indicated they were complete, a second screen prompted them to add any final details they might have missed in their initial recall. This approach during practice helped participants to provide additional information they would have otherwise not reported and recognize the depth of recall we were expecting[63–66]. They were instructed to provide a full recall of all details remembered during the main experimental procedures.

Once participants demonstrated an understanding of the tasks, they proceeded to the main experiment. Participants read and marked event boundaries for each of the three narratives, one narrative at a time. Unlike the practice task, participants were instructed to focus only on the large

event units. The free-recall phase immediately followed each narrative segmentation phase. They were encouraged to provide exhaustive detail, but unlike the practice task, they were only provided one opportunity for recall. The order in which the three narratives were presented was counterbalanced across participants.

In what follows, we first present methods and results for the automated event segmentation approach and subsequently the methods and results for the automated recall scoring.

## Automated event segmentation

To assess the efficacy and generalizability of our approach, we implemented the automated segmentation method using both OpenAI's GPT-4[19] and Meta AI's LLaMA 3.0[20].

**LLM segmentation procedure.** The three narrative texts were separately input into GPT-4 and LLaMA 3.0 through their respective application processing interface (API) using Python 3.11.5[67]. A zero-shot prompt[68] as model input was used as described previously using GPT-3[1]. Instructions were purposefully vague and were constructed to simulate the instructions provided to participants[12,13]:

*An event is an ongoing coherent situation. The following story needs to be copied and segmented into large events. Copy the following story word-for-word and start a new line whenever one event ends, and another begins. This is the story*:

A full narrative was then inserted, followed by additional text to refresh and reiterate the instructions:

*This is a word-for-word copy of the same story that is segmented into large event units*:

The temperature parameter in modern large language models, including GPT and LLaMA, determines the randomness of the model outputs. Temperature values range between 0 and 1, where higher values make responses more random, while lower values are more focused and deterministic. Previous work focused solely on a temperature 0 implementation[1]. To provide a more comprehensive investigation of the consistency of event boundary identification, we ran the model separately across three temperature values: 0, 0.5, and 1. Although OpenAI's API allows a true zero temperature, generating text based on the highest probability (i.e., greedy decoding), the LLaMA implementation requires a strictly positive temperature, as it generates text by sampling from a probability distribution. We used 0.1, the lowest permissible value, which is effectively deterministic. This value is functionally equivalent to a temperature of 0, and we refer to it as such throughout for consistency with GPT-4. It is worth noting that a temperature of 0, while generally considered deterministic, may still exhibit variability due to stochastic factors within the API—such as nondeterministic behaviours in token sampling or caching[69]. Nonetheless, temperature 0 (or 0.1 for LLaMA) provides functionally deterministic outputs and is suitable for assessing model consistency. At the same time, a temperature of 0 may be too rigid and miss salient event boundaries that are evident to humans. By varying the model temperature, we allow the model to generate more variable outputs, helping us test whether it could identify a broad range of meaningful event boundaries while still maintaining overall consistency. To avoid incomplete responses, we also set the max_tokens parameter to 4096. This parameter defines the maximum number of tokens —units of text, such as words or punctuations—that the model can generate. A high allocation ensures that the model can produce a segmentation response for the entire narrative. This approach ultimately helped us assess both the stability and flexibility of the LLM segmentation behaviours.

For each of the three narratives, two LLMs, and three temperature conditions, we ran the model 20 times such that the sample size was equivalent to that of the human participants. We refer to one of the 20 LLM runs of the model as an LLM instance (i.e., an *instance* mirrors a participant). This repetition also allowed us to record and quantify the variability in segmentation responses across model instances. After the text was segmented, the text was tokenized, and the event boundary locations were recorded based on their word number within the text.

## A. Agreement Index

**Binary word-level series for each participant**

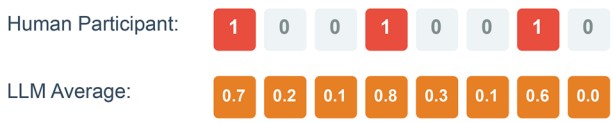

Participant 1: `1 0 0 1 0 0 1 0`

Participant 2: `1 0 0 0 1 0 1 0`

Average (others): `0.9 0.1 0.0 0.5 0.8 0.0 0.9 0.0`

Agreement Index = *corr*(participant, average of others)

## B. Human-to-LLM Agreement

Human Participant: `1 0 0 1 0 0 1 0`

LLM Average: `0.7 0.2 0.1 0.8 0.3 0.1 0.6 0.0`

Human-LLM Agreement = *corr*(participant, LLM average)

## C. Shared vs. Distinct Boundaries

🟩 Shared Boundary (Human + LLM)  🟧 Distinct Boundary (Human only)

Human Average: `1 0 0 0.8 0 0 0.2 0`

LLM Average: `1 0 0 0.7 0 0 0 0`

$Amplitude_{shared}$ vs. $Amplitude_{distinct}$

## D. Between-Group Consistency

**100 permutations**

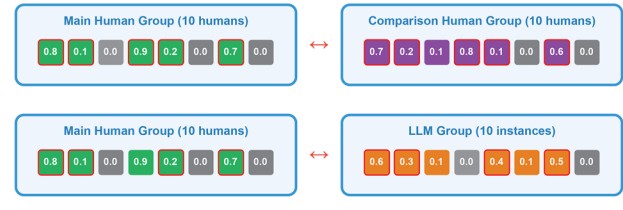

| Main Human Group (10 humans) | ↔ | Comparison Human Group (10 humans) |
| Main Human Group (10 humans) | ↔ | LLM Group (10 instances) |

Consistency = $P_{matching}$

**Fig. 2 | Analytical methods. A** Agreement index correlates each participant's binary word-level boundary series against the average of all other participants in the same group. The amplitude of the average word-level series represents the proportion of participants identifying an event boundary at that location. **B** Human-to-LLM agreement correlates a single human participant with the average LLM word-level series. **C** Shared vs. distinct boundaries categorize boundaries as identified by both groups (green) or humans only (orange). The proportion of human participants (i.e., amplitude of the average word-level series) is compared between shared and distinct boundary conditions. **D** Between-group consistency is computed using a permutation test across 100 iterations. Humans are randomly split into two groups of 10, and a separate group of LLM instances is sampled. Consistency is measured as the proportion of event boundaries in the main human group—regardless of amplitude— that are also found in the comparison group (i.e., second human group, LLM group). Red-outlined boxes indicate locations where both groups marked an event boundary.

### Analysis of event segmentation data

General statistical methodology. Statistical analyses were performed using R 4.4.0[70]. Linear mixed effects models were conducted using *lme4*[71] and interpreted using ANOVA tables through *lmerTest*[72]. Linear mixed effects models were generally applied when dependent variables were measured at either a ratio or interval scale and the assumption of independence was met. Residual distributions were inspected to verify approximate normality and homoscedasticity and were therefore assumed to satisfy these assumptions, but no formal tests were conducted, as such models are robust to moderate deviations from normality[73–75]. Effect sizes were generated using the *effectsize* package[76]. Post hoc analyses were conducted using *emmeans*[77] with Tukey adjusted *p*-values (denoted as $p_T$) and Kenward–Roger approximated degrees of freedom[78]. Exact *p*-values are reported when possible and approximated to common conventions when values are subject to computational underflow (i.e., $p < 0.0001$).

The random effects structure in the linear mixed effects models was tailored to the design of each analysis. For within-subject analyses, we included random intercepts for subject and narrative to account for variability at both levels. For between-subject analyses, random intercepts were included only for the narrative. This is noted and justified in the corresponding methods sections. Although a maximal random structure was initially tested, which was justified by the design, nearly all models resulted in singular fits. As a result, we adopted a simplified structure with random intercepts only (i.e., removal of random slopes) to ensure model stability and comparability across models[79,80]. Specific analytical methods are outlined in their respective sections below.

**Number of event boundaries.** For each human participant and LLM instance (separately for different models and temperatures), the number of identified boundaries was counted for each narrative. To account for differences in narrative length, the number of identified boundaries was divided by the word count of the corresponding narrative and scaled to a per-1000-words basis to standardize the comparisons. A linear mixed effects model was calculated using the scaled number of events as the dependent variable and segmentation condition (human, LLM temperature 0, 0.5, 1) as the between-subjects factor with a random effect of narrative.

**Segmentation agreement index.** To analyze the degree to which human participants and LLM instances agreed in their placement of event boundaries, we calculated a segmentation agreement index[8,10,14,62] (Fig. 2A). Each narrative text was tokenized into individual words, and for each participant and LLM instance (separately for GPT-4 and LLaMA 3.0 and temperatures), a binary classification was applied: a word was assigned a value of one if an event boundary had been identified just prior to it, and a zero otherwise[5,10,14]. This process resulted in a word-level series for each narrative, participant, LLM, and temperature.

We employed a leave-one-out approach, calculating the point-biserial correlation between the participant's word-level series and the averaged word-level series across all other participants[8,10,14,15,62]. Similarly, we calculated an agreement index for each LLM instance using the same leave-one-out approach, separately for each LLM and temperature condition. A linear mixed effects model was conducted with the agreement index as the dependent variable and segmentation condition (i.e., human, LLM temperature 0, 0.5, 1) as the independent measure, with narrative as a random effect.

**Agreement between LLM and human event boundaries.** We used a similar agreement index approach to assess alignment between LLM instances and human participants (Fig. 2B). For each narrative, we

created a word-level series for each participant and calculated an agreement index between each participant's word-level series and the average word-level series of the LLM, separately for each model and three temperature conditions. A linear mixed effects model was used to assess the within-subject effect of temperature condition. Narrative and participant were included as random factors.

**Proportion of participant responses at LLM-human shared vs. distinct boundaries**. Event boundaries identified by a greater proportion of participants may be more salient[13,81]. To evaluate whether LLMs can detect the most salient human event boundaries, we first generated a word-level series for each narrative across all human participants and for each LLM temperature condition. Human identified boundaries were classified as *shared boundaries* if at least one LLM instance identified a boundary at the same location, or as *distinct boundaries* if no LLM instances marked a boundary at that location (Fig. 2C). We then compared the proportion of human participants (i.e., boundary amplitude from average word level series) identifying shared versus distinct boundaries using a linear-mixed effects model, with temperature, boundary type (i.e., shared, distinct), and their interaction as fixed effects and narrative included as a random effect. Each boundary's participant proportion score was treated as a separate data point in the model.

**Between-group consistency**. Despite the common agreement in boundary placement across participants[3,10,12,13,81], not every participant identifies the same event boundaries[14]. We assessed how consistently two groups of humans identify event boundaries compared to how consistently a group of humans and a group of LLM instances identify boundaries (Fig. 2C). To this end, we used the word-level series for each participant and LLM instance and calculated a permutation test across 100 iterations. For each iteration (and separately for each narrative), we randomly split the 20 human participants into two groups of 10, designating one as the main group and the other as the comparison group. Similarly, we randomly selected 10 LLM instances from the total pool of 20 LLM instances and labeled this group as the LLM comparison group. We then calculated the average word-level series for each group and identified all peaks in these averaged series as event boundaries.

The between-group consistency was evaluated in two ways. First, we calculated the proportion of event boundaries for which both the main human group and the human comparison group identified the same boundaries. Second, we calculated the proportion of event boundaries for which both the main human group and the LLM comparison group identified the same boundaries. Since the average word-level series often did not produce the same number of event boundaries across groups, the proportions were calculated based on the sample with the fewest number of events. The effects of segmentation group (human, 0, 0.5, 1) on the proportion of matching event boundaries were assessed through a linear mixed effects model. The dependent variable was the proportion of matching event boundaries for each iteration, with segmentation group as a fixed effect and narrative included as a random effect. Each iteration of the permutation test contributed a data point to the model, resulting in, for each narrative, 100 proportion scores per segmentation group.

**Human ratings of normative boundaries procedure**
A subsequent behavioural experiment was conducted to investigate how humans agree with the event boundaries identified by LLMs ($N = 11$, see demographics above). Although LLaMA 3.0 was initially considered due to its open-source and privacy-preserving advantages, it was not included in these subsequent analyses because its segmentation performance was notably poorer than GPT-4. Given these limitations, we focused on GPT-4 with a temperature of 0, which yielded the most consistent and human-aligned results (described below). Normative boundaries from GPT-4 for each narrative were identified as the highest $n$ number of boundaries, where $n$ is the mean number of event boundaries across the 20 instances of the model[14]. All identified normative boundaries were located at the end of sentences.

Participants were given the same three written narrative texts as in the main experiment, except that event boundaries identified by GPT-4 were already marked on the text printout of each narrative. As a control condition, we also included an identical mark at non-boundary locations (i.e., approximately event centres) for each narrative. The event boundaries and non-boundaries were marked as red lines between two sentences in the text. There were no differences in appearance between the boundary types (boundary, non-boundary), and all markings were located at the end of sentences. Each participant received the same set of marked texts, but the order of narrative presentation was counterbalanced. Participants were tasked to read through the narratives and to indicate whether each marking was a true event boundary or a non-boundary. They also rated the confidence in their decision on a scale from 1 to 10, with 1 indicating low confidence and 10 indicating high confidence. To ensure that participants fully understood the task, they first completed a practice task with the same 400-word narrative as in the original experiment. Instructions were given both verbally and in written form to ensure clarity.

For the analysis, the confidence rating for markings that participants thought were not event boundaries was negative-coded. Subsequently, the ratings were linearly scaled from 1 to 10 to a range of −1 to 1, where −1 represents high confidence that a marking was not an event boundary, 1 represents high confidence that a marking was an event boundary, and 0 represents low confidence for both cases. For each participant, we averaged the scaled confidence rating at the boundary and non-boundary points across narratives. One-sample *t*-tests were then performed for each boundary condition to assess whether the average confidence significantly differed from zero. An independent samples t-test was conducted to compare the average confidence ratings between boundary and non-boundary points.

**Automated recall assessment**
The automated event segmentation approach was used as the basis to automate recall scoring, which is described in this section. This approach leverages the concept that human memory is structured around discrete events[5,34,35].

**Recall assessment procedure**
Recall transcripts and event segmentation. Recall data from the original experiment were used for this analysis. Full datasets from two participants and data for one narrative from another participant were removed from this analysis due to an error during recording. Recall audio was initially transcribed through Otter.ai[82], and a manual transcriber verified and cleaned all transcripts. For each narrative text and recall text, normative boundaries were identified using the automated segmentation approach with OpenAI's GPT-4 model with a temperature parameter of 0. In line with the results above, GPT-4 alone was used in these recall assessment methods since its event boundary responses evidently outperformed that of LLaMA 3.0. This resulted in narrative text segments and recall text segments that were used for recall scoring.

**Sematic representations**. The current approach capitalizes on semantic representations of text that are derived from modern LLMs. To demonstrate the effectiveness and generalizability of the automated scoring approach, we employed the Universal Sentence Encoder (USE, v4)[56], OpenAI Embeddings (text-embedding-3-large)[19], Language-Agnostic BERT sentence embedding (LaBSE)[57], and Masked and Permuted Language Modelling (MPNet, all-mpnet-base-v2)[58]. These models encode text into high-dimensional vectors called text-embeddings, where vectors for semantically similar texts (e.g., phone and computer) show a higher correlation than vectors for texts that are less semantically similar (e.g., eye and street). By correlating the vectors for narrative and recall text segments, we can capture their semantic similarity[37,83–85]. For the current analysis, each narrative text segment and each participant's recall

segments were encoded into an embedding vector, separately for each model: USE, OpenAI, LaBSE, and MPNet. This process yielded one vector per narrative text segment and recall text segment.

### Analysis of recall data

Intersubject agreement. To investigate the extent to which meaningful information is represented in recall data (independent of the relation to the narratives), we assessed the similarity among participant's recall using an intersubject correlation approach[36,86,87]. For each participant, a correlation matrix was calculated, using Spearman correlation, between the embedding vectors of their recall and the embedding vectors of another participant's recall of the same narrative. The matrix was resized into a square matrix[88,89] using the *Scikit-image* package[90] relative to the number of narrative events to enable averaging and standard analysis across participants. Spearman correlation was used over cosine similarity because it is less sensitive to outliers[91–93].

A correlation matrix was calculated for each $N-1$ participant combination. Correlation values along the diagonal of this correlation matrix reflect the degree to which the participant recalled the narrative in the same order as that of another participant, whereas correlation values along the reverse diagonal reflect the degree to which the participant recalled the narrative in a reverse order (control condition). For each participant, we extracted and averaged the diagonal values from their correlation matrices, resulting in $N-1$ values per participant. This procedure was repeated for the reverse diagonal elements. These scores were then averaged separately to generate a single diagonal and reverse diagonal score per participant. Through the diagonal and reverse diagonal correlations, we can assess the temporal alignment of participants' recall, reflecting a shared narrative structure, and thus whether there is meaningful information extracted by the text-embedding vectors. A linear mixed effects model was applied to predict intersubject agreement as a function of embedding model (USE, OpenAI, LaBSE, MPNet) and score type (original vs. reverse order) with participants and narrative included as random effects.

Recall scoring. For each participant and narrative, a semantic similarity (correlation) matrix was created[37,42] by calculating the Spearman correlations between the text-embeddings of each narrative segment and the text-embeddings of each recall segment. To account for differences in the number of narrative and recall events, the matrix was transformed to a square correlation matrix relative to the number of narrative events[88,89], using the *Scikit-image* package[90]. This resulted in one narrative × recall matrix for each participant and narrative. For each narrative event (i.e., row) in the correlation matrix, we identified the maximum correlation value, representing the best-matching recalled event. The maximum correlation served as an event-specific recall score for each narrative event.

To quantify narrative recall performance and the efficacy of the automated scoring approach, we computed the average recall score for each participant across all narrative events for the automated recall scores. This produced a single narrative recall score for each participant and narrative. To establish the reliability of recall, we compared these scores to a baseline condition, where the embeddings of the recall were correlated with the embeddings from the two unrelated narratives, and the maximum score per narrative event was extracted and averaged across events.

For the statistical analysis, we calculated a linear mixed effects model with the embedding model (USE, OpenAI, LaBSE, MPNet) and score type (actual recall score, random recall score) and their interaction as fixed effects, as well as subject and narrative as random effects. Prior to analysis, scores were grouped by model and standardized (z-transformed).

Split-half consistency between automated and human rater scores. Two trained research assistants were tasked to rate the relationship between automated recall scores and human judgment. Unlike the AI-based approach, which calculates recall scores by comparing pre-segmented narrative and recall texts, we provided human raters with the segmented narrative but the full, unsegmented recall text. This decision was made to align with prior human scoring methods, where raters assess recall based on predetermined narrative targets rather than pre-segmented recall excerpts[38,40,41]. By allowing raters to evaluate gist recall on a scale from 0 to 10—where 0 indicates no mention of the narrative event and 10 indicates near-verbatim recall—we ensured that their judgments reflected an overall understanding of the narrative content rather than a strict, detail-by-detail match. This approach maintains comparability with previous gist-based rating systems[38,40,94] and resembles most the semantic similarity analysis of the automated scoring compared to other manual scoring approaches.

To measure the consistency between the automated recall scores and the human raters, we conducted a split-half consistency analysis[95–97]. The participant pool was divided randomly into two equal groups. The first group was used to calculate the Spearman correlation between the automated recall scores and human raters (concatenating all recall scores from participants), while the second group served as a control, whereby the recall scores were shuffled to compute a null distribution representing a random correlation. This process was repeated across 10,000 iterations, resulting in 10,000 actual correlation values and 10,000 random correlation values. The significance was determined using a one-tailed nonparametric permutation test, comparing the average correlation value with the 10,000 shuffled correlation values[97]. A significant correlation would indicate that the automated recall scores correlated with the human recall scores at a rate better than chance. To obtain an overall correlation value between automated recall scores and human rater scores while accounting for the reduced sample size for this analysis, we performed the Spearman-Brown split-half reliability correction (denoted as $\rho_{SB}$) on the average correlation value obtained from the distribution[97].

Standardized regression between automated and human rater scores. To directly assess the relationship between human scores and automated scores among different text-embedding models, the human rater scores and automated event recall scores were standardized within each text-embedding model. A linear mixed-effects analysis was conducted for each AI model to evaluate how well automated recall scores predict human ratings, with participant and narrative included as random effects to account for individual differences and narrative-specific variability. Because both predictor and outcome variables were standardized, the resulting beta coefficients represent standardized effect sizes, indicating the strength of the relationship in terms of standard deviations. This allows for intuitive interpretation of effect sizes and direct comparison of the predictive strength across different models.

### Reporting summary

Further information on research design is available in the Nature Portfolio Reporting Summary linked to this article.

## Results

### Automated event segmentation

Number of event boundaries. For GPT, the temperature 1 condition identified significantly more events relative to temperature 0 ($t_{234} = 5.20$, $p_T = 2.64 \times 10^{-6}$), temperature 0.5 ($t_{234} = 5.58$, $p_T = 2.76 \times 10^{-7}$), as well as relative to human participants ($t_{234} = 3.93$, $p_T = 6.43 \times 10^{-4}$; main effect of segmentation condition: $F_{3,234} = 13.19$, $p = 5.49 \times 10^{-8}$, $\eta_P^2 = 0.14$, 95% CI [0.07, 0.22]; Fig. 3B). There were no significant differences between the temperature 0, 0.5, nor humans (largest $t_{234} = 1.72$, $p_T = 0.315$). For LLaMA 3.0, the model identified more events than human participants for all temperature conditions (smallest $t_{234} = 5.83$, $p_T = 1.09 \times 10^{-7}$, main effect of segmentation condition: $F_{3,234} = 22.38$, $p = 8.96 \times 10^{-13}$, $\eta_P^2 = 0.22$, 95% CI [0.07, 0.22]; Fig. 3D), but there were no significant differences between the different temperatures. Hence, LLaMA and GPT only with temperature 1 appear to overestimate the number of event boundaries.

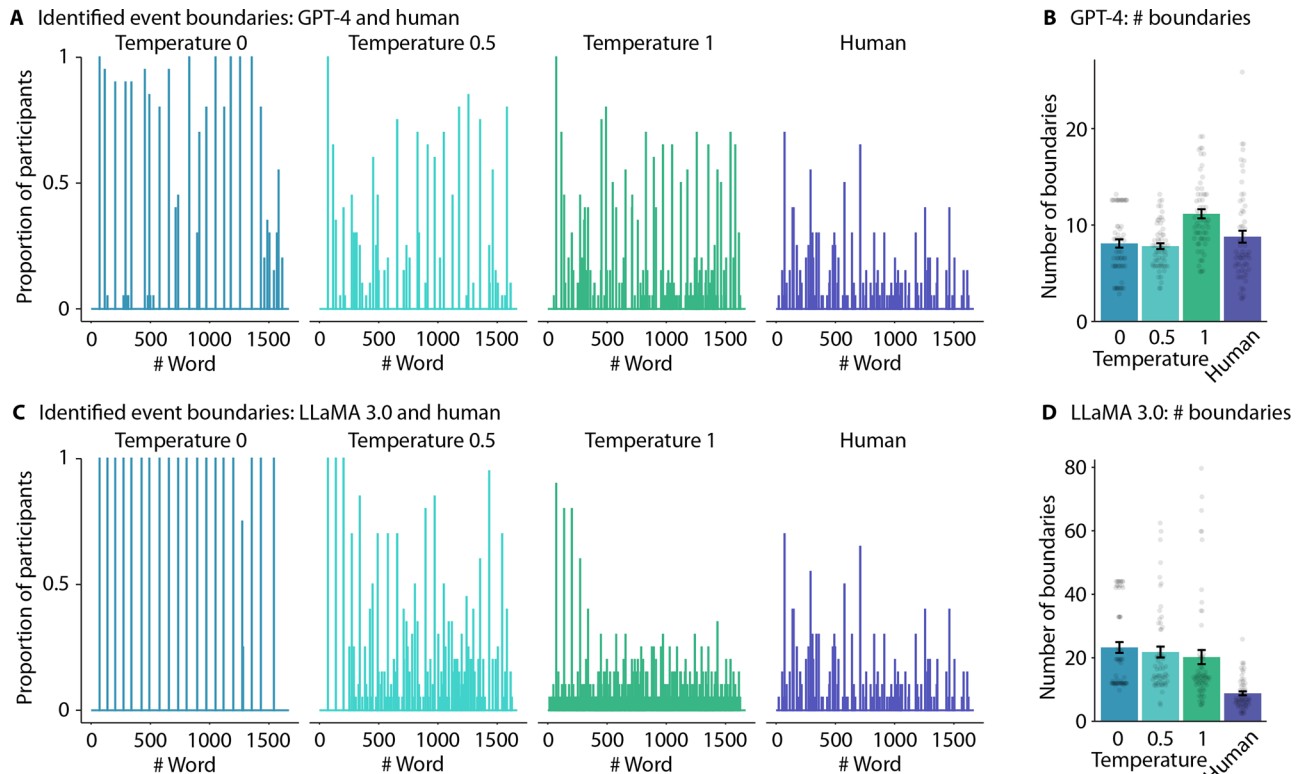

**Fig. 3 | Segmentation results. A** Location of identified event boundaries in Narrative One for GPT. Each pane shows the proportion of participants (n = 20 humans) or LLM instances (n = 20 model instances, per temperature) that identified event boundaries for each word of the narrative. **B** Number of identified event boundaries. Bars show the mean across participants per 1000 words. **C** Location of identified event boundaries in Narrative One for LLaMA. **D** Number of LLaMA identified event boundaries per 1000 words. Error bars represent ±1 SEM across participants (n = 20 human, per narrative) and across independent model runs (n = 20 model instances, per temperature, per narrative).

**Segmentation agreement index.** For GPT, the agreement index (i.e., how well did a participant's or LLM instance's boundary placement agree with the rest of the group) was greatest for the temperature 0 condition amongst all other conditions (smallest $t_{234} = 8.43$, $p_T < 0.0001$, main effect of segmentation condition: $F_{3,236} = 226.36$, $p = 3.78 \times 10^{-69}$, $\eta_p^2 = 0.74$, 95% CI [0.69, 0.78]; Fig. 4A). The agreement index was greater for temperature 0.5 than temperature 1 ($t_{234} = 12.52$, $p_T < 0.0001$) and humans ($t_{234} = 14.09$, $p_T < 0.0001$; temperature 1 vs. humans, $t_{234} = 1.83$, $p_T = 0.26$). For LLaMA, the temperature 1 condition produced the lowest agreement index than all other conditions (smallest $t_{234} = 9.10$, $p_T < 0.0001$, main effect of segmentation condition: $F_{3,234} = 206.51$, $p = 1.84 \times 10^{-65}$, $\eta_p^2 = 0.73$, 95% CI [0.67, 0.77]; Fig. 4B), whereas temperature 0 produced the greatest agreement index (smallest $t_{234} = 16.541$, $p_T < 0.0001$). The temperature 0.5 model resulted in agreement that did not statistically differ from the agreement of the human participants ($t_{234} = 0.47$, $p_T = 0.96$). These results suggest that responses using temperature 0 best reflect non-random narrative specific segmentation behaviours.

**Agreement between human and LLM event boundaries.** We further assessed how the human event boundaries aligned with the LLM event boundaries using the agreement index. For GPT, humans alignment did not significantly differ between the temperature 0 and temperature 0.5 conditions ($t_{156} = 0.28$, $p_T = 0.96$, main effect of temperature: $F_{2,156} = 24.15$, $p = 7.31 \times 10^{-11}$, $\eta_p^2 = 0.24$, 95% CI [0.13, 0.34]; Fig. 4A), whereas the alignment to the temperature 1 condition was reduced (smallest $t_{156} = 6.24$, $p_T < 0.0001$). For LLaMA, humans were best aligned to the temperature 0 condition (smallest $t_{156} = 4.533$, $p_T = 3.41 \times 10^{-13}$, main effect of temperature: $F_{2,156} = 29.65$, $p = 1.22 \times 10^{-11}$, $\eta_p^2 = 0.28$, 95% CI [0.16, 0.38]; Fig. 4B), but humans were still better aligned to the

temperature 0.5 than 1 condition ($t_{156} = 3.12$, $p_T = 6.01 \times 10^{-3}$). Using the human agreement index as a reference (Fig. 4A), these results suggest that individual human participants are generally more aligned with GPT-4 responses for the 0 and 0.5 temperatures than responses of other human participants; however, this evaluation does not seem to hold for LLaMA 3.0 (Fig. 4B).

**Proportion of participant responses at LLM-human shared vs. distinct boundaries.** We subsequently assessed the average proportion of human participants who identified event boundaries when they matched (i.e., shared) and did not match (i.e., distinct) the LLM-generated event boundaries. Indeed, for both models (GPT and LLaMA), the proportion of participants for shared event boundaries was greater than the proportion of participants at distinct locations (GPT: $F_{2,742} = 330.02$, $p = 2.68 \times 10^{-61}$, $\eta_p^2 = 0.31$, 95% CI [0.26, 0.36]; Fig. 4C; LLaMA: $F_{1,404} = 91.43$, $p = 1.16 \times 10^{-20}$, $\eta_p^2 = 0.18$, 95% CI [0.12, 0.25]; Fig. 4D). This proportion also decreased as the temperature increased (GPT: $F_{1,742} = 16.98$, $p = 6.18 \times 10^{-8}$, $\eta_p^2 = 0.04$, 95% CI [0.02, 0.07]; LLaMA: $F_{2,743} = 4.08$, $p = 1.73 \times 10^{-2}$, $\eta_p^2 = 0.01$, 95% CI [0.00, 0.03]). The results suggest that LLMs typically identify the most salient human event boundaries.

**Between-group consistency.** Splitting participant and LLM instance groups in half enabled us to evaluate the consistency across groups. For GPT at temperatures 0 and 0.5, the proportion of the identified event boundaries that aligned between a human group and a GPT group was greater than the aligned event boundaries between two human groups (smallest $t_{1194} = 13.58$, $p_T < .0001$, main effect of segmentation condition: $F_{3,1194} = 672.37$, $p = 6.90 \times 10^{-256}$, $\eta_p^2 = 0.63$, 95% CI [0.60, 0.66]; Fig. 4B); however, this pattern did not hold temperature 1 ($t_{1194} = 20.78$,

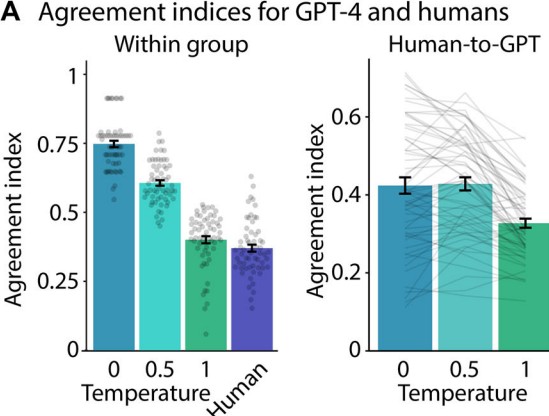

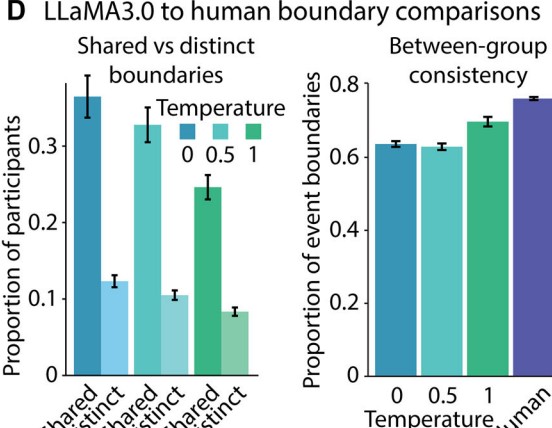

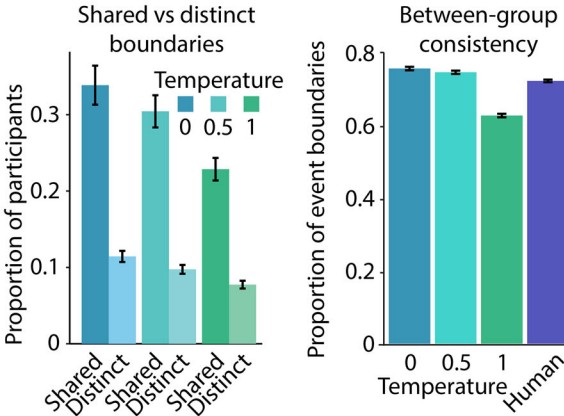

**Fig. 4 | Segmentation agreement. A** GPT event boundary agreement and alignment to human responses. **B** LLaMA event boundary agreement and alignment to human responses. **C** Human-to-GPT event boundary comparison. **D** Human-to-LLaMA event boundary comparisons. Error bars represent ±1 SEM across participants ($n$ = 20 humans, per narrative) and across independent model runs ($n$ = 20 model instances, per temperature, per narrative).

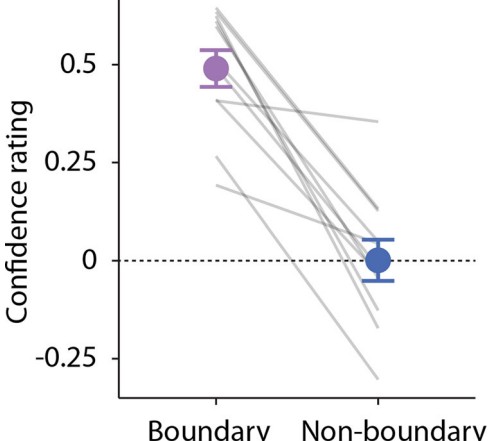

**Fig. 5 | Results of humans rating GPT event boundaries and non-boundaries.** Averaged participant confidence ratings at boundary and non-boundary conditions ($n$ = 11). Error bars represent ±1 SEM across participants.

$p_T < .0001$), where the two human groups were better aligned than humans with GPT. For LLaMA, in contrast, the human responses among their own group consistently had the highest proportion of aligned event boundaries relative to all temperature conditions (smallest $t_{1194} = 8.01$, $p_T < 0.0001$, main effect of segmentation condition: $F_{3,1194} = 129.88$, $p = 8.24 \times 10^{-73}$, $\eta_p^2 = 0.25$, 95% CI [0.21, 0.28]; Fig. 4D). In other words,

GPT more consistently produced event boundaries that aligned with human responses, and more so than humans among themselves.

**Human ratings of GPT-identified boundaries.** A separate group of participants that rated whether they agreed or disagreed with GPT (temperature 0) identified event boundaries and non-boundaries (i.e., event centres) revealed a greater confidence for the boundary than non-boundary condition ($t_{10} = 7.28$, $p = 2.65 \times 10^{-5}$, $d = 2.20$, 95% CI [1.07, 3.30]; Fig. 5). The confidence rating for event boundaries were significantly >0 ($t_{10} = 10.46$, $p = 1.05 \times 10^{-6}$, $d = 3.16$, 95% CI [1.66, 4.63]), indicating that participants correctly identified the GPT-identified normative event boundaries, whereas the confidence rating did not statistically differ from zero for the non-boundaries ($t_{10} = 0.02$, $p = 0.98$, $d = 6.67 \times 10^{-3}$, 95% CI [−0.58, 0.60]). These results provide additional justification that GPT can correctly identify true normative event boundaries that align with human responses.

**Summary.** The results suggest that LLMs, particularly GPT-4, identified boundaries similar to those identified by human participants: (i) major event boundaries identified by humans were also identified by LLMs; (ii) LLMs relative to humans were as consistent, if not more, in identifying boundaries compared to humans among themselves; and (iii) a separate group of humans were confident about GPTs boundary placement. More deterministic parameters (i.e., lower temperatures) aligned better with human responses. GPT-4 produced segmentation results that both aligned better with human responses and produced more consistent results across model

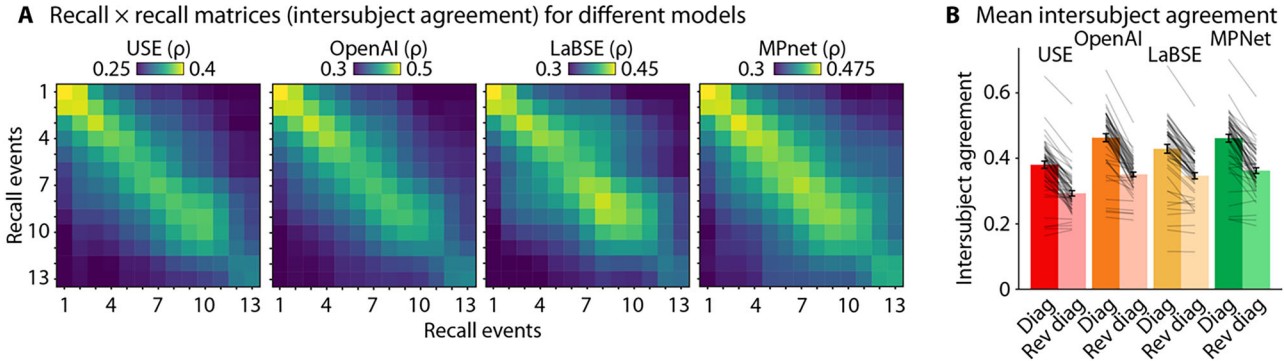

**Fig. 6 | Intersubject agreement. A** Averaged recall × recall correlation matrices across the three narratives. Segmented text was embedded, and a correlation matrix was calculated representing the semantic similarity between recall events between different participants. To visualize the average intersubject agreement matrices across narratives with different numbers of events, the matrices were resized to the median number of narrative events. **B** Intersubject agreement ($n = 20$ participants, per narrative) across the correlation matrix diagonal (Diag) was compared to the intersubject agreement across the reverse diagonal (Rev diag). Larger diagonal scores than reverse diagonal scores mean that narratives were more consistently recalled across participants. Error bars represent ±1 SEM across participants.

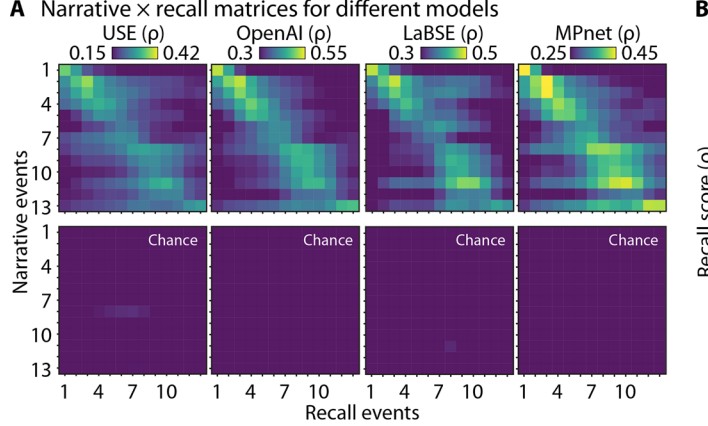
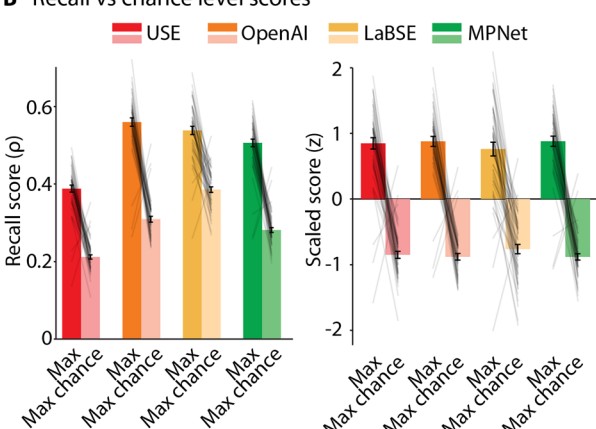

**Fig. 7 | Event recall analysis. A** Narrative × recall matrices and chance level matrices (derived from non-corresponding narratives). For visualization of the average matrices across narratives, matrices were transformed to square matrices based on the median number of narrative events (13 × 13). Resizing was performed *after* computing recall similarity scores for each narrative and does not affect the original analysis. **B** Narrative recall scores were averaged to obtain a single narrative recall score per participant ($n = 20$ participants, per narrative). Recall scores were calculated for both corresponding and non-corresponding narratives (control). The bar plots on the right show the same data as on the left, but $z$-transformed to visualize the similar magnitude of the effect across LLMs. Error bars represent ±1 SEM across participants.

responses compared to LLaMA 3.0. GPT-4 with temperature 0 may thus be recommended for future use.

### Automated recall assessment

**Intersubject agreement.** We first assessed the extent of agreement across the recall of different participants. Figure 6A shows the average correlation matrix that reflects the semantic similarity of participants' recall for individual narrative events. The values along the diagonal of the recall × recall correlation matrix reflect the temporal agreement among participants' recall, whereas the reverse diagonal reflects the extent to which participants recalled the narrative in the reversed order relative to other participants (used as a control). Across all four embedding models, the intersubject agreement across the matrix diagonal was greater than agreement for the reverse order ($F_{1,428} = 336.76$, $p = 6.56 \times 10^{-56}$, $\eta_P^2 = 0.44$, 95% CI [0.38, 0.50]: Fig. 6B), indicating that recall temporality was significantly above chance levels across participants. The USE produced the lowest intersubject agreement scores compared to all other models (smallest $t_{428} = 6.91$, $p_T < 0.0001$, main effect of model: $F_{3,428} = 43.49$, $p = 1.50 \times 10^{-24}$, $\eta_P^2 = 0.23$, 95% CI [0.17, 0.30]). The results indicate that participants indeed recall narratives in a similar

temporal order, thus providing evidence that the correlation matrices comprise meaningful information.

**Assessment of recall accuracy.** We examined the sensitivity of automating narrative recall scoring. All text-embedding models produced narrative recall scores that were better than a baseline condition when recall was evaluated against non-corresponding narratives ($F_{1,397} = 1253.55$, $p = 6.53 \times 10^{-125}$, $\eta_P^2 = 0.76$, 95% CI [0.72, 0.79]; Fig. 7B). Effects were similar across embedding models for standardized recall scores ($F_{1,397} = 1.40$, $p = 0.24$, $\eta_P^2 = 0.01$, 95% CI [0.00, 0.03]), but for unstandardized model outputs the overall magnitudes and difference to the control condition depended on the model used ($F_{1,399} = 163.66$, $p = 3.62 \times 10^{-69}$, $\eta_P^2 = 0.55$, 95% CI [0.49, 0.60]).

**Split-half consistency between automated scores and scores from human raters.** We assessed the relationship of automated recall scores and human gist ratings through a split-half consistency analysis (Fig. 8A). We observed significant consistency for all freely available text embedding models (USE: $\rho_{SB} = 0.52$, $p < .0001$; LaBSE: $\rho_{SB} = 0.62$, $p < .0001$; MPNet: $\rho_{SB} = 0.52$, $p < .0001$), as well as the proprietary model (OpenAI: $\rho_{SB} = 0.64$, $p < .0001$), demonstrating that the automated approach

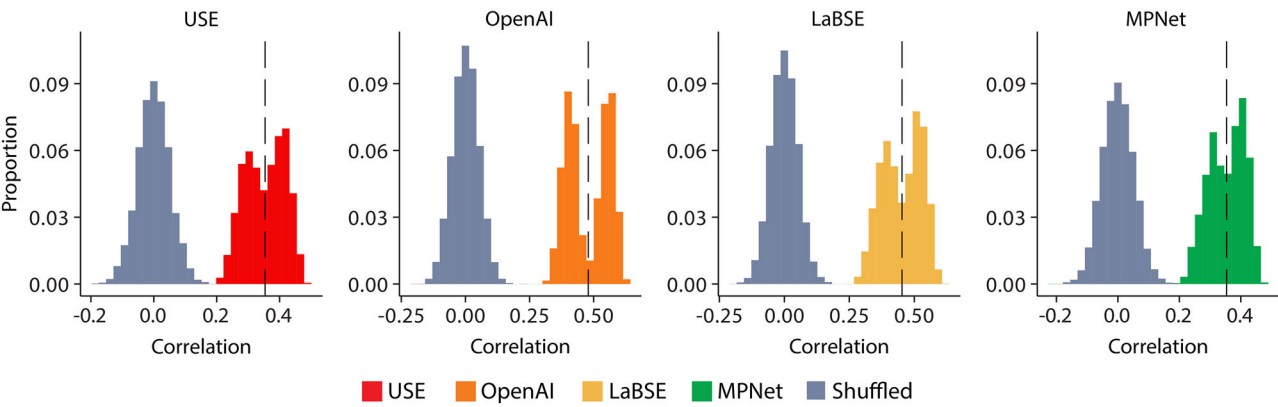

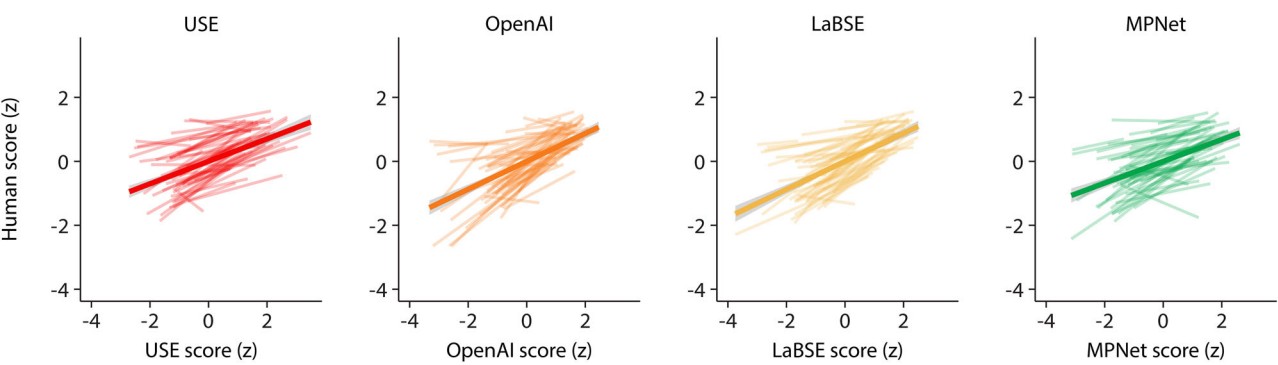

**Fig. 8 | Event recall analysis. A** Split-half correlation analysis over permuted 10,000 iterations assessing the correlation of model to human scores ($n = 20$ participants). **B** Standardized regressions of model scores to human scores ($n = 20$ participants). Shaded error ribbons represent ±1 SEM across participants.

captures relevant recall features that are also captured in human recall scoring.

A linear mixed effects model was subsequently conducted to further assess the relationship between human rater scores and automated recall scores for each model individually. Prior to analysis, event recall scores were standardized. The analysis revealed a significant positive relationship between the human rater scores and automated recall scores produced by the USE ($\beta = 0.37$, $t_{758} = 11.66$, $p = 5.03 \times 10^{-29}$), OpenAI ($\beta = 0.52$, $t_{758} = 15.74$, $p = 1.47 \times 10^{-48}$), LaBSE ($\beta = 0.43$, $t_{758} = 12.81$, $p = 3.48 \times 10^{-34}$), and MPNet ($\beta = 0.36$, $t_{758} = 10.84$, $p = 1.40 \times 10^{-25}$). Collectively, these findings suggest that all four models predict human rater scores (Fig. 8B).

**Summary**. Across different text-embedding models (USE, OpenAI, LaBSE, and MPNet), the results show that automating narrative recall assessments produces meaningful recall scores. Although various text-embedding models are trained differently and have distinct embedding sizes, performance was largely consistent. Each model could effectively score event recall like humans at a rate better than chance. Based on these results, we recommend the LaBSE model as it is free to use and best captures nuance in narrative and recall structures and is language independent.

## Discussion

The current study investigated whether LLMs can be used to automate event segmentation and recall assessments. We leveraged chat-completion models (GPT-4 and LLaMA 3.0) for event segmentation, alongside multiple text-embedding models—Universal Sentence Encoder (USE), OpenAI Embeddings, Language-Agnostic BERT Sentence Embedding (LaBSE), and Masked and Permuted Network (MPNet)—for semantic similarity analysis

in recall assessments. The results demonstrate the effectiveness of LLMs in aligning with human judgement of event boundaries and effectively assessing recall ability, providing a scalable, cost-effective approach to investigating event perception and subsequent memory.

### Event segmentation

Segmenting the continuous environments of everyday life into meaningful events is crucial for shaping episodic memory and future recall[5,9,33,34]. By accurately capturing event boundaries, automated LLM-based event segmentation can serve as a valuable proxy for understanding perceptual processes. Knowing where event boundaries occur in stimulus materials can also be useful for the development of experimental methods or generation of stimuli that aim to manipulate memory[98–100]. For the latter purpose, automating event segmentation may be particularly powerful, and the current study shows that LLMs can approximate human-identified event boundaries.

Previous work has used a prompt-based segmentation approach with GPT 3.0[1], which included multiple model runs to assess reliability; however, each run was treated independently, focusing on the alignment of individual runs and human annotations, and evaluating the consistency of this effect across several single-run instances. In contrast, our approach evaluates how a group of model runs collectively relates to human responses by treating each model run as an individual instance and allowing us to mirror the structure of our human analyses. Although the previous work relied primarily on responses from a single run of GPT-3.0 with a fixed temperature setting of 0 under the assumption that the model would produce highly reliable and deterministic outputs, it did not explicitly quantify within-model consistency or variability across iterations. By assessing the consistency across model runs, we were able to show that LLMs more consistently identify event boundaries than humans, and critically, that humans

agree more consistently with LLM-identified boundaries than among humans themselves. LLMs thus appear to reliably identify event boundaries.

The current analyses also assessed how model outputs change with varying levels of randomness and highlighted temperature values between 0 and 0.5 that emulate human perception. Lower temperatures resulted in more deterministic and human-aligned segmentation outputs; however, we did observe some variability across instances, even in the lowest temperature conditions. This aligns with recent work showing that LLMs can exhibit nondeterministic behaviours due to token sampling and API-level processing[69], indicating that although variability is minor, exact replicability cannot be assumed. Obtaining normative event boundaries from the average responses across LLM instances may be advantageous. Higher temperatures generated more event boundaries, but these boundaries were more scattered and led to reduced agreement between LLM instances and human segmentation. Humans also produced event boundaries with noticeable variability, but it still appears that temperature zero captures human responses the best, with more consistency than humans had amongst themselves.

One important distinction to previous work lies in the modality of participant data for which we make the comparisons[1]. collected human data such that participants were exposed to auditory stimuli, which likely may not align with the way LLMs process and segment text-based inputs. In our experiment, participants read the same textual stimuli used for model segmentation, providing a more direct comparison between human and LLM performance, which is crucial for validating the method before extending it to other modalities, such as spoken language, where additional factors may influence segmentation. By first establishing a reliable text-based benchmark, we create a foundation for future work exploring multimodal contexts.

The current research focused on GPT-4 and LLaMA 3.0 because they are currently the flagship LLMs, and both can serve as tools for segmentation, each with distinct advantages. GPT-4 showed consistently higher agreement with human boundaries than LLaMA 3.0 (Fig. 4), but GPT-4 is associated with fees, whereas LLaMA weights are publicly available. Costs can be a potential barrier, although user fees for newer versions of GPT-4o have reduced substantially (openai.com/api/pricing), possibly making the investment for higher accuracy worthwhile. Other up-and-coming models, such as DeepSeek v3[101,102] are even cheaper suggesting a potential trend in increased affordability of high-performance models, which could broaden overall accessibility.

Although LLaMA evades potential privacy concerns associated with proprietary models because it can be run locally, newer LLaMA models, such as LLaMA 3.1 70B, are associated with substantial hardware costs and memory requirements, as it has become too large even for a powerful computer with a high-grade graphics card to manage. Despite using a high-end consumer system (32 GB RAM and an NVIDIA RTX 4080 GPU with 16 GB VRAM), we were unable to run the 70B parameter LLaMA 3.1 locally. Furthermore, computations of high-complexity models like GPT-4 can have significant environmental carbon impacts[103,104]. Unlike models that are deployed locally, reliance on frequent API calls can increase energy usage, especially if redundancy is introduced through backoff strategies for handling rate limits. Instead, researchers could consider batching tasks or pre-processing data in a manner that reduces the frequency of API requests.

Although GPT-4 provided higher alignment with human boundaries, making it suitable for high-stakes applications, LLaMA 3.0 offers robust performance that may remain suitable for research contexts, particularly where scalability, cost, and ethical considerations are prioritized. Other models with transformer-based architectures, such as BLOOM[105], Gemini[106], Claude[107], and most recently DeepSeek[101] may offer comparable segmentation performance to GPT-4 and LLaMA, providing researchers with a range of viable alternatives.

## Recall assessments

We used the automated event segmentation approach and leveraged a variety of text-embedding models to automate recall scoring. We showed that the recall among different participants was shared, showing that there is

meaningful information extracted by the text-embedding vectors used to automate recall assessments (Fig. 6). Participants similarly recalled narratives in the original temporal order, which is consistent with previous work showing that individuals recall materials in the order in which they perceived them[33,36,42,45,108,109] and highlights the sensitivity of the current approach. We further showed that automated scores predict human ratings (Fig. 8), suggesting that this approach can be used instead of manual human scoring. Ultimately, these results demonstrate the efficacy of using text-embedding models in extracting narrative-specific semantic relationships to assess narrative recall.

A few other recent works have developed approaches to automate recall scoring[42,43,46–49]. These methods rely on advanced analytical methods such as topic modelling, Hidden Markov Models, and fine-tuning of existing large language models, often requiring, at a minimum, a moderate computational background to implement. While some of these recent techniques have achieved near-perfect correlations to manual raters ($r \approx 0.99$), they have primarily used short narrative passages (~60 words)[46,48], where scores based on constrained outputs and a predetermined set of details rather than overall gist[43,45,46,48]. While this approach allows for a precise assessment of specific elements, it may not fully capture how memory operates for longer naturalistic narratives.

Methods such as topic modelling and HMM have also been successfully applied[42]; however, their aims differ from the current approach. Specifically, such methods focus on modeling latent thematic transitions over time, whereas our study is explicitly grounded in the theoretical relationship between event segmentation and recall. Additionally, some studies have focused on recall at the clausal level[47,49], which, while useful for sentence-level recall, may not align with how larger narrative structures are encoded in and recalled from memory. In contrast, the present study assesses recall of longer narratives (~ 1500 words), allowing for an examination of everyday memory processes. Given that real-world memory retrieval often involves reconstructing overarching themes and event structures rather than isolated clauses, the current approach may be particularly useful for assessing recall of longer narratives. Our observed standardized coefficients ($\beta = 0.37$–$0.52$), derived from linear-mixed effects models, are inherently more conservative than Pearson correlations typically reported in prior work, and thus are appropriately comparable to estimates from other work using similar open-ended recall tasks (e.g., $r \approx 0.60$)[43]. Critically, our findings demonstrate that out-of-the-box LLMs can effectively score recall without extensive fine-tuning, making implementation more accessible and scalable for broader research applications.

The text-embedding models used for the current study (i.e., USE, OpenAI, LaBSE, MPNet) are lightweight and optimized for accessibility. They are small enough to be loaded and used quickly, making them ideal for researchers with limited computational resources or for real-time applications. Despite their efficiency, they provide robust sentence-level semantic representations, enabling accurate assessments of narrative recall. Based on our results, there were no distinct advantages of using the proprietary OpenAI embeddings over the freely available alternatives (USE, LaBSE, MPNet; Fig. 8), further enhancing the accessibility of the automated approach, as research can leverage cost-effective models without sacrificing accuracy. Like the segmentation approaches, the free models can be run locally rather than requiring API calls to OpenAI, erasing potential privacy concerns. Among the freely available models, LaBSE stands out as a particularly strong model for this procedure, exhibiting the highest consistency ($\rho_{SB} = 0.62$; Fig. 8A) with human recall scores and a significant predictive relationship ($\beta = 0.43$; Fig. 8B). LaBSE is further designed for multilingual applications by enabling consistent semantic encoding across languages[57,86]. These advantages make LaBSE a powerful, cost-effective alternative to proprietary models like OpenAI text-embeddings, while ensuring comparable performance.

## Limitations

Although the current study demonstrates the feasibility of using large language models for automating event segmentation and recall scoring, its

scope was limited to a small number of models. As generative AI systems continue to evolve rapidly, differences in architecture and training data may lead to variations in performance, reproducibility, and potentially, interpretability over time. As previously mentioned, this approach also carries inherent environmental and financial costs associated with large-scale model computation, which mainly constrain accessibility for some research contexts. Additionally, because model outputs are inherently non-deterministic, identical inputs can yield slightly different segmentation estimates, though averaging across multiple model instances may help mitigate this variability. Finally, our automated recall assessment approach focuses on semantic similarity and gist-based scoring rather than verbatim recall or exact detail counts. While this aligns well with many research questions about episodic memory and narrative comprehension, some research questions may require fine-grained detail analyses that would need to be supplemented with additional methods to capture verbatim accuracy and detail retention.

## Conclusion

The current study evaluated the applications of large language models (LLMs) for automated event segmentation and recall assessments of written narratives. By leveraging both chat-completion models (GPT-4, LLaMA 3.0) and various text-embedding models (USE, LaBSE, OpenAI, and MPNet), we show that LLMs can replicate human segmentation patterns and provide reliable recall assessments. Our findings highlight the importance of temperature settings in model outputs, with lower temperatures yielding the most consistent alignment with human judgment. GPT-4 exhibited superior segmentation alignment within humans compared to LLaMA 3.0. Moreover, semantic similarity analyses with LLMs enabled robust assessments of the temporal structure of recall and recall accuracy, as well as alignment with human raters. These results suggest that LLMs offer an accessible, scalable, and cost-effective solution for research on event perception and memory as well as clinical applications.

## Data availability

All data and materials generated or analyzed in this study are available via GitHub at github.com/ryanapanela/EventRecall.

## Code availability

Analytical code, computational workflows, and the accompanying *EventRecall* module implementing the automated event segmentation and recall scoring procedure described are available via GitHub at github.com/ryanapanela/EventRecall. The exact version of this repository corresponding to the analyses reported in this manuscript and the initial release of the module is archived on Zendo[110].

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

## Acknowledgements

We thank Nicholas Wong and Xiaoning Wang for their computational expertise; Tiffany Lao for her help with data collection; and Sarah Bobbitt, Saba Junaid, and Andrew Cole for their help with manual transcribing and recall scoring. This research was supported by the Natural Sciences and Engineering Research Council of Canada (Discovery Grant: RGPIN-2021-02602), Canadian Institutes of Health Research (Funding Reference Number—R.A.P.: 193310, B.H.: 195994), and the Canada Research Chairs Program (CRC-2023-00383). The funders had no role in study design, data collection and analysis, decision to publish, nor preparation of the manuscript.

## Author contributions

R.A.P.: Conceptualization, methodology, software, formal analysis, investigation, data curation, writing—original draft, writing—review and editing, visualization, project administration. A.J.B.: Conceptualization, methodology, resources, writing—review and editing, supervision. M.D.B.: Conceptualization, methodology, writing—review and editing, and supervision. B.H.: Conceptualization, methodology, formal analysis, writing—original draft, writing—review and editing, visualization, supervision, project administration, and funding acquisition.

## Competing interests

The authors declare no competing interests.
