## [Transparent Peer Review file · Communications Psychology]

Event Segmentation Applications in Large Language Model Enabled Automated Recall Assessments

Corresponding Author: Mr Ryan Panela

Version 0:

Decision Letter:

Dear Mr Panela,

Thank you for your patience during the peer-review process. Your manuscript titled "Large Language Model Applications for Event Segmentation in Automated Recall Assessments" has now been seen by 4 reviewers, whose comments are appended below. They find your work of interest but raised some important points. We are interested in the possibility of publishing your study in Communications Psychology, but would like to consider your responses to these concerns and assess a revised manuscript before we make a final decision on publication.

We therefore invite you to revise and resubmit your manuscript, along with a point-by-point response to the reviewers. Please highlight all changes in the manuscript text file.

Editorially, we consider it important that the revised manuscript provides comparisons to other models/methods of event segmentation as requested by Reviewers 1 and 2. Please also provide an accessible version of your method as requested by Reviewer 2 and addressing all concerns of Reviewer 3.

I am attaching a checklist that details critical reporting requirements for the revised manuscript. Please attend to each item and ensure your manuscript is fully compliant. We are requesting that your manuscript aligns with these requirements as this facilitates the evaluation of your manuscript, reducing delays in re-review and potential future acceptance. If your revised manuscript is not aligned with these requests on major issues, such as those concerning statistics, it may be returned to you for further revisions without re-review. Additional information can be found in our style and formatting guide Communications Psychology formatting guide.

If the revision process takes significantly longer than four months, we will be happy to reconsider your paper at a later date, provided it still presents a significant contribution to the literature at that stage.

Please use the following link to submit your

- revised manuscript,
- point-by-point response to the referees' comments,
- cover letter (as a separate document),
- the Editorial Policy Checklist (see below),
- the Reporting Summary (see below), and
- the completed Editorial Request Table (attached):

Link Redacted

Thank you for the opportunity to review your work.

Best regards,

Jennifer Bellingtier

on behalf of
Jesse Rissman, PhD
Editorial Board Member
Communications Psychology
orcid.org/0000-0001-8889-5539

REVIEWER EXPERTISE:

Reviewer #1 LLMs, event segmentation
Reviewer #2 computational modeling
Reviewer #3 code review

REVIEWER REPORTS:

Reviewer #1 (Remarks to the Author):

In Panella et al., the authors investigate whether large language models can be used to automate two core components of naturalistic memory research: event segmentation and recall scoring. Using GPT-4 and LLaMA 3.0, they evaluate how well LLM-generated segmentations align with human boundaries, and they test a range of embedding models for scoring free recall transcripts against annotated recall units. The study contributes a practical pipeline for automating these tasks and offers a useful resource for researchers interested in scalable alternatives to manual annotation. Below are some concerns with the current version of the manuscript that could be addressed in a revised version of the manuscript.

1) Many parts of the manuscript reflect replications of existing work. For instance, the segmentation component builds directly on Michelmann et al., with temperature-based sampling and a comparison of different LLMs as the main extension and the recall scoring analyses replicate and extend prior work by Heusser et al. and Raccah et al. These extensions provide a valuable comparison that will likely be of practical use to others in the field. That said, both components lean more toward empirical validation than conceptual or methodological innovation. It would be helpful to clarify more explicitly in the manuscript, what parts reflect replication and extension of prior work and what contributions are unique to this manuscript. In this context: A key contribution of the manuscript seems to be the connection of an event segmentation pipeline with the scoring of free recall. Could the authors elaborate more on how these two parts connect and potentially position the study's novel contributions based on the relationship between their pipeline and earlier work?

2) The rationale for including temperature as a variable in the segmentation task could be further clarified. While the manuscript explains (in Line 221) that temperature controls output randomness, its conceptual connection to event segmentation was not clear on reading the manuscript. Could the authors explain what the conceptual purpose of increasing randomness in the segmentation is? Specifically, what does the randomness afford besides making models and humans less aligned (which is arguably an undesired outcome with event segmentation and potentially any norming task)

3) In Line 274, the paper introduces "shared" boundaries (defined as any overlap between GPT-4 and at least one human) and "distinct" boundaries (no overlap across all instances). Could this definition have led to some of the results where the authors observe higher agreement rates on those boundaries?

4) In Line 324, the authors introduce a normalization based on the minimal and maximal point-biserial correlation. While this measure captures sensitivity to shared boundaries, it was not clear how specificity is afforded by the method. Specifically, a rater (or model) that only places a single event boundary at a point that overlaps with the rater (or model) of comparison would obtain an adjusted correlation of 1, which is the maximum possible correlation that can be obtained from a single response. Could the authors compare their measure with other measures that measure sensitivity and specificity of the segmentation? Relatedly, what is the null distribution of random segmentation?

5) The current study uses a single fixed prompt throughout the manuscript, so the authors should not claim to test "prompt-level variability" but rather temperature-based variability or sampling consistency.

Some minor comments include:

1) In the introduction, LLaMA is presented as a key open-source and privacy-preserving alternative to GPT-4, which helps motivate the study. However, it is then largely dropped from the main analyses—with limited explanation—being excluded from the human validation and recall scoring components. The brief mention that its performance was "notably poorer" may benefit from further elaboration. Given its framing in the introduction, it would be helpful to more clearly justify its exclusion from later stages and to clarify its intended role in the study. If the goal is to generalize findings across LLMs, providing more detail on LLaMA's performance and limitations would strengthen that claim.

2) In Line 563, the manuscript notes that prior work did not account for model variability across prompts; however, the cited study (Michelmann et al.) did include multiple runs.

3) Line 140, The manuscript refers to introducing "new analysis metrics," but it is somewhat unclear which elements are novel contributions versus adaptations of prior methods.

- 4) There are some instances where the wording could be more precise. For instance, several findings are framed in terms of “LLMs” more broadly, though the analyses rely only on GPT-4. In another instance, in Line 388, the manuscript refers to “true boundaries,” which might be more accurately described as “GPT-identified boundaries.”
- 5) In Figure 3B, the title appears to reference LLaMA, but the plot title is “Human-to-GPT”. This may be a labeling error.
- 6) The y-axes across subplots in Figure 3 use different scales, which makes it harder to visually compare effects across conditions. Standardizing the y-axis range across these plots would improve readability and support more intuitive comparisons.
- 7) In Figure 3C, I was interested in understanding why the proportions of shared and distinct boundaries do not sum to one. A brief note on how these values are computed mathematically would help readers interpret the plot.
- 8) In Line 432 and Figure 5, the manuscript mentions that recall and narrative matrices are resized to the median number of segments for visualization. It would be helpful to clarify whether this resizing occurs before or after computing the recall similarity scores.
- 9) In Line 590, the manuscript notes that “LLaMA 3.1 [models] are associated with substantial hardware costs... too large even for a powerful computer.” It would be helpful to specify which model size this refers to. For example, the 8B version of LLaMA 3.1 can typically run on a high-end consumer machine, and the authors may be referring to the 70B model. Since consumer hardware can differ, it would also be more informative to specify the exact hardware requirements directly.
- 10) Typos
 - Line 294: repeated use of “for which.”
 - Line 362: “his” should read “this.”
 - Figure 4 title: “conference rating” should be “confidence rating.”
 - Line 398: “be recommend” should be “be recommended.”

Reviewer #2 (Remarks to the Author):

The manuscript describes an approach to segmenting human recall of events using LLMs, rather than traditional approaches that rely on human raters. They then validate this model by applying it to human recall data.

First, I should note that this is not my domain of expertise. The methods are interesting and I broadly support publication, but I cannot comment on how this work fits in with the literature space (either related methodological approaches or theoretical aspects of event-related memory). However the manuscript is well-written and I understand the core technical aspects sufficiently. Below, I have listed several points that I think could improve the manuscript.

1) In describing the segmentation metrics (p. 8-12), it is easy to get lost, in part because the metrics are described in natural language. It would help the reader to include equations (or perhaps visualizations) to clarify each metric’s calculation and how they are all distinct (some of them sound quite similar at first!). This is in contrast to the results, which are much easier to digest at a glance because the authors have provided many easy to understand figures.

2) Most analyses make use of mixed-effect models, but the random effect structure could be clarified. First, a glance at the code on github suggests the authors primarily use crossed-intercept models, but do not include random slopes (this isn’t clear in the text). Best practice is to use a maximal model justified by the design (Barr et al., 2013) and iteratively reduce this structure when encountering singular fit or convergence issues. Random intercept only models are known to inflate Type I errors (which is probably why they are so common, despite not being best practice). The authors should revise or justify these choices. Second, the authors should justify any discrepancies between models. There are some cases where they use two random intercepts (narrative and participant) and others where they use only one. It might be obvious why to the authors, but in trying to make sense of it for the first time it is a bit confusing. The authors could provide these justifications in each section, or – since many of the analyses parallel each other – include a preliminary paragraph about the mixed effect design and then note where (if) analyses deviate from this.

Barr, D. J., Levy, R., Scheepers, C., & Tily, H. J. (2013). Random effects structure for confirmatory hypothesis testing: Keep it maximal. *Journal of memory and language*, 68(3), 255-278.

3) In the introduction, the authors generally talk about “perception” and “spatiotemporal information”. They should make clear early on (and in the abstract) that their methods are related to text or transcribed speech data. I assumed that might be the case, but was not too confident until I reached the Methods. There are plenty of experiments in which participants recall narratives after watching videos, but the methods described would ignore any non-text data and probably wouldn’t be suitable.

4) The motivation for the work could be strengthened.

4a) First, the authors point out that there are several other methods to accomplish the same task using HMMs, topic models, and at least one that uses LLMs. But no comparisons to these models are made. Can these models be applied to the same data or not? Are they worse than the authors’ method? The authors claim they are harder to implement, but that’s all.

4b) Second, the authors frequently refer to this as an “automated” approach (including in the title) but I don’t really see how. Sure, I can copy the prompt to do event segmentation – easy enough. But for recall assessment, I need to programmatically use a language model to encode responses, create a correlation matrix, etc. Essentially, I need to reproduce the author’s

analysis pipeline. To their credit, they have their code on github, and it is fairly clear, but hosting code doesn't make it automated. When I hear "automated" what I expect is a tool where researchers can plug in a .csv of their data (and possibly an OpenAI token) and it spits out results that can be analyzed in the standard fashion. My experience in authoring an open-source methods tool in my own area has taught me that if you actually want people to *use* your method, you have to make it dead easy – including for those who study event memory but know little about LLMs and are not methodologists. Otherwise, people will mostly only cite your paper in the way that the authors cite previous HHM and topic model approaches – something that has been done in the past, not a method that is being currently used. For example, the authors could cobble together a bunch of functions from their existing code into a single .R file (and from there, it is easy enough to host it as a library on CRAN!) I think this would be a boon for the authors as well, who would surely get more citations if their methods are actually used.

(Relatedly, the authors state that "manual scoring guidelines and approaches may differ between research groups possibly leading to inconsistencies in the literature and reproducibility challenges." Proposing yet *another* method does not seem like it really solves that problem.)

This is perhaps the most time-consuming critique of mine to address, but I think it is also the most central for a methods paper where the goal certainly feels like convincing your readers to use this method. I encourage the authors to strongly consider it. If the goal is only to see whether LLMs segment data like humans, the paper needs to be reframed - but I find that framing much less appealing.

5) The analyses at temperature 0 are confusing. I typically assume that a temperature parameter of 0 in softmax means deterministic. They even use this word when referring to Michelmann et al.'s approach on p. 22. But from the figures, the results are clearly not deterministic. In Fig 2A, there is clearly variability in GPT-4. In Fig 2C, LLaMA is *almost* deterministic, though there appears to be one event boundary that not all instances agreed on. Relatedly, I would expect the within-group agreement for temperature 0 to be perfect (in fact, I questioned why the authors needed to run it 20 times). But the results in Fig 3 reveal that it is not. It would help to clarify why this is the case.

6) (p. 19+) The split-half reliabilities are fairly low, but the LLM/human correlations are noticeably lower. Is this an issue? In simple linear regression, standardized betas are the same as Pearson correlation (the authors include random effects here so I'm not so sure how that affects the interpretation). Here, they are reported in the range of [.36, .52]. So, approximately, R^2 in the range of [.13, .27]. Fine – but not amazing. I will leave it to other reviewers who know more about the field, but in absence of any comparisons to other methods cited, it's hard to tell whether I should be impressed by this. Perhaps the authors could comment on it.

Minor:

p. 8 the authors state that higher temperature values are "more creative". This is a strong take, I would remove.

P. 8 Authors should clarify what "max_tokens" is in plain language

Fig 3B Should "Human-to-GPT" be "Human-to-LLaMA" ?

Fig 7B I suggest adding either an identity line or at least rescaling the axes to be the same. Correlations are important, but absolute agreement is important too, and it's harder to discern from these figures with different axes.

Reviewer #3 (Remarks to the Author):

My task was to review the code and ensure that it runs and is documented well.

README

There isn't enough information in the README.md file. The only information is:

""

This repository contains data and files used for:

Panela, R. A., Barnett, A. J., Barense, M. D., & Herrmann, B. (2025). Event Segmentation Applications in Large Language Model Enabled Automated Recall Assessments (Version 1). arXiv. <https://doi.org/10.48550/ARXIV.2502.13349>

""

A README.md file should explain what's in the repository. It should explain that there are both R and Python files. It should state what output they both produce, and it should explain how to install the dependencies and run them.

For R using RStudio automatically prompts and installs packages. This is not always the case for Python. It is expected that a requirements.txt file or some other means of tool/library management is outlined. To produce a requirements.txt files, the following can be run on the command line if a virtual environment was used and the libraries/packages were installed using pip

pip freeze > requirements.txt

See below for the output under "# Python code" .

It would be good to outline in the README what data is included and what data isn't -- I could produce types of graphs, but not all the graphs in the paper. This could possibly be because not all the data was included. If all the data was indeed included, it wasn't clear to me how to produce all the outputs and not the types of outputs.

R code

When the data files are used in the code it would be good to have some signposting such as "PATH_TO/" to remind the user to insert the absolute path to the files or a variable could be defined so the user would only have to change the path to the data file in one place.

In the data file "gpt_segmentation_data.csv", the column "word_number" is referred to in the file "gpt_segmenation_analysis.Rmd" on line 64, after the data is ingested. However, there is no "word_number" column, only a "word_numbers" column in the .csv data file. Perhaps this could be corrected.

In the data file "visual_time_word_data.csv", the column "temperature" is referred to in the file "gpt_segmenation_analysis.Rmd" on line 157, after the data has been ingested. However, there is no "temperature" column in that .csv file at all. Therefore the code between lines 153 to 302 could not be run at all. If somehow the column "temperature" had to be adjoined to the data file, this was not clear.

In gpt_segmentation_analysis.Rmd, the code after line 302 could be run, but because the variable "temperature" (see above) is used after this point. However, the "Identified event boundaries" graphs (Figure 2 Segmentations in the paper) do not look like the graphs that I could reproduce (because previous segments of the code could not be run as mentioned above).

The R code files are not as well commented as the Python code is. There are headings for each of the sections, but no explanation of the code as there are in the Python files.

Python code

In the README file, it would be good to know if the Python code is expected to be run first, or the R code or whether it doesn't matter.

Additionally, it would be good to state which Python file to run. One file (segmentation_functions.py) serves as resources/tools file, while the file the "gpt_llama_segmentation.py" file is the main file to be executed.

There is an error in the definition for "get_finish_reason". It is defined on line 164 as the following with 2 arguments:

```
get_finish_reason(responses, choice):  
...
```

However when it is called on line 237, it has 1 argument.

```
get_finish_reason(item)
```

Therefore the Python code threw the following error:

```
TypeError: get_finish_reason() missing 1 required positional argument: 'choice'
```

Once I had fixed this in the code, I reran it.

As mentioned above, it would be good what dependancies are required for the Python code with their required version because this is one way to ensure the longevity of the code. My output when I run "pip freeze > requirements.txt" after installing all the dependencies was:

```
""  
distro==1.9.0  
exceptiongroup==1.2.2  
filelock==3.18.0  
fonttools==4.57.0  
fsspec==2025.3.2  
h11==0.14.0  
httpcore==1.0.8
```

```
httpx==0.28.1
huggingface-hub==0.30.2
idna==3.10
Jinja2==3.1.6
jiter==0.9.0
kiwisolver==1.4.8
MarkupSafe==3.0.2
matplotlib==3.10.1
mpmath==1.3.0
networkx==3.4.2
numpy==2.2.5
openai==1.76.0
packaging==25.0
pandas==2.2.3
pillow==11.2.1
psutil==7.0.0
pydantic==2.11.3
pydantic_core==2.33.1
pyparsing==3.2.3
python-dateutil==2.9.0.post0
pytz==2025.2
PyYAML==6.0.2
regex==2024.11.6
requests==2.32.3
safetensors==0.5.3
scipy==1.15.2
seaborn==0.13.2
six==1.17.0
sniffio==1.3.1
sympy==1.13.3
tokenizers==0.21.1
torch==2.7.0
tqdm==4.67.1
transformers==4.51.3
typing-inspection==0.4.0
typing_extensions==4.13.2
tzdata==2025.2
urllib3==2.4.0
"""
```

After installing the required libraries and changing the file paths, I was still not able to run the Python code because of the following error:

ValueError: `temperature` (=0) has to be a strictly positive float, otherwise your next token scores will be invalid.

Full output from raised error is pasted here:

```
"""
Traceback (most recent call last):
File "/Users/misticam/Projects/event-recall/code/segmentation/produce_segmentation/gpt_llama_segmentation.py", line 10,
in <module>
llama_events = llama_segmentation(f'/Users/misticam/Projects/event-recall/data/stories/{story}.txt', iters=iterations,
temperature=temp)
File "../event-recall/code/segmentation/produce_segmentation/segmentation_functions.py", line 284, in
llama_segmentation
curr_response = prompt_llama(curr_prompt, temperature=temperature)
File "/Users/misticam/Projects/event-recall/code/segmentation/produce_segmentation/segmentation_functions.py", line 134,
in prompt_llama
response = pipe(prompt_message,
File "/Users/misticam/Projects/event-recall/envs/review/lib/python3.10/site-
packages/transformers/pipelines/text_generation.py", line 280, in __call__
return super().__call__(Chat(text_inputs), **kwargs)
File "../event-recall/envs/review/lib/python3.10/site-packages/transformers/pipelines/base.py", line 1379, in __call__
return self.run_single(inputs, preprocess_params, forward_params, postprocess_params)
File "../event-recall/envs/review/lib/python3.10/site-packages/transformers/pipelines/base.py", line 1386, in run_single
model_outputs = self.forward(model_inputs, **forward_params)
File "../event-recall/envs/review/lib/python3.10/site-packages/transformers/pipelines/base.py", line 1286, in forward
model_outputs = self._forward(model_inputs, **forward_params)
File "../event-recall/envs/review/lib/python3.10/site-packages/transformers/pipelines/text_generation.py", line 385, in
```

```
_forward
output = self.model.generate(input_ids=input_ids, attention_mask=attention_mask, **generate_kwargs)
File ".../event-recall/envs/review/lib/python3.10/site-packages/torch/utils/_contextlib.py", line 116, in decorate_context
return func(*args, **kwargs)
File ".../event-recall/envs/review/lib/python3.10/site-packages/transformers/generation/utils.py", line 2358, in generate
prepared_logits_processor = self._get_logits_processor(
File ".../event-recall/envs/review/lib/python3.10/site-packages/transformers/generation/utils.py", line 1194, in
_get_logits_processor
processors.append(TemperatureLogitsWarper(generation_config.temperature))
File ".../event-recall/envs/review/lib/python3.10/site-packages/transformers/generation/logits_process.py", line 282, in
_init_
raise ValueError(except_msg)
ValueError: `temperature` (=0) has to be a strictly positive float, otherwise your next token scores will be invalid.
'''
```

In the paper "temperature=0" is one of the hyperparameters that is shown, so there are some discrepancies. The above ellipses "..." in the pasted error simply are my local file paths.

I am happy to rerun the code when it gets updated. I think this repo would be a great resource for other researchers once the issues are resolved.

Reviewer #4 (Remarks to the Author):

I co-reviewed this manuscript with one of the reviewers who provided the listed reports.

EDITORIAL POLICIES

We ask that you ensure your manuscript complies with our editorial policies and reporting requirements.

To that end, we require revised manuscripts to be accompanied by two completed items: a reporting summary that collects information on study design and procedure, and an editorial policy checklist that verifies compliance with all required editorial policies

- <https://www.nature.com/documents/nr-reporting-summary.zip>>Nature Research Reporting Summary
- <https://www.nature.com/documents/nr-editorial-policy-checklist.pdf>>Editorial Policy Checklist

All points on the policy checklist must be addressed. Your revised manuscript can only be sent back to the referees if these checklists are completed and uploaded with the revision.

Notes: If you have submitted a Stage 1 Registered Report, Review, Primer, Comment, or Perspective you do not need to submit these forms. If you have already submitted these forms, you may disregard this request.

** Visit Nature Research's author and referees' website at <http://www.nature.com/authors>>www.nature.com/authors for information about policies, services and author benefits**

Version 1:

Decision Letter:

Dear Mr Panela,

Your manuscript titled "Event Segmentation Applications in Large Language Model Enabled Automated Recall Assessments" has now been seen by our reviewers, whose comments appear below. In light of their advice I am delighted to say that we are happy, in principle, to publish a suitably revised version in Communications Psychology.

We therefore invite you to revise your paper one last time to address the remaining concerns of our reviewers (especially regarding the code concerns of Reviewer 3) and a list of editorial requests. At the same time we ask that you edit your manuscript to comply with our format requirements and to maximise the accessibility and therefore the impact of your work.

EDITORIAL REQUESTS:

SUBMISSION INFORMATION:

OPEN ACCESS:

*** TRANSPARENT PEER REVIEW:** Communications Psychology uses a transparent peer review system. On author request, confidential information and data can be removed from the published reviewer reports and rebuttal letters prior to publication. If you are concerned about the release of confidential data, please let us know specifically what information you would like to have removed. Please note that we cannot incorporate redactions for any other reasons.

*** CODE AVAILABILITY:** All Communications Psychology manuscripts must include a section titled "Code Availability" at the end of the methods section. We require that the custom analysis code supporting your conclusions is made available in a publicly accessible repository at this stage; please choose a repository that generates a digital object identifier (DOI) for the code; the link to the repository and the DOI must be included in the Code Availability statement. Publication as Supplementary Information will not suffice.

* DATA AVAILABILITY:

Link Redacted

**** This url links to your confidential home page and associated information about manuscripts you may have submitted or be**

reviewing for us. If you wish to forward this email to co-authors, please delete the link to your homepage first **

Best regards,

Jennifer Bellingtier

Jennifer Bellingtier, PhD
Senior Editor
Communications Psychology

Jesse Rissman, PhD
Editorial Board Member
Communications Psychology
orcid.org/0000-0001-8889-5539

REVIEWER EXPERTISE:

Reviewer #1 LLMs, event segmentation
Reviewer #2 computational modeling
Reviewer #3 code review

REVIEWERS' COMMENTS:

Reviewer #1 (Remarks to the Author):

The authors have addressed all of my concerns in their revision.

Reviewer #2 (Remarks to the Author):

I would like to thank the authors for their attentiveness to the points raised by reviewers. I have no major concerns remaining.

Reviewer #3 (Remarks to the Author):

My part of the review was to ensure that the code runs smoothly. The follow up task was to ensure that the Python code ran. The installation instructions don't work for me as expected.

I ran the command from the README:

```
pip install git+https://github.com/ryanapanela/EventRecall.git
```

I created a script to run with the first code snippet under the heading "Python API Usage" and got the following error:

```
ModuleNotFoundError: No module named 'segmentation'
```

I then used tried to install via "setup.py" by running:

```
python setup.py build  
python setup.py install
```

And still got the same error.

I installed the dependencies in requirements.txt and ran the code snippet again. When I got it running I got the error:

```
FileNotFoundError: [Errno 2] No such file or directory: 'test/Run.txt'
```

The file Run.txt is in the folder data/stories/. Once I moved the file to a folder named "test" in the same folder as my script, the could get the code snippet from the README running.

The automatic installation packaging doesn't work, but the core code runs once you add your own API key.

It would be good to know which version of Python is required in the README.

Reviewer #4 (Remarks to the Author):

I co-reviewed this manuscript with one of the reviewers who provided the listed reports. This is part of the Communications Psychology initiative to facilitate training in peer review and to provide appropriate recognition for Early Career Researchers who co-review manuscripts.

REVIEWER #1

In Panela et al., the authors investigate whether large language models can be used to automate two core components of naturalistic memory research: event segmentation and recall scoring. Using GPT-4 and LLaMA 3.0, they evaluate how well LLM-generated segmentations align with human boundaries, and they test a range of embedding models for scoring free recall transcripts against annotated recall units. The study contributes a practical pipeline for automating these tasks and offers a useful resource for researchers interested in scalable alternatives to manual annotation. Below are some concerns with the current version of the manuscript that could be addressed in a revised version of the manuscript.

1. Many parts of the manuscript reflect replications of existing work. For instance, the segmentation component builds directly on Michelmann et al., with temperature-based sampling and a comparison of different LLMs as the main extension and the recall scoring analyses replicate and extend prior work by Heusser et al. and Raccach et al. These extensions provide a valuable comparison that will likely be of practical use to others in the field. That said, both components lean more toward empirical validation than conceptual or methodological innovation. It would be helpful to clarify more explicitly in the manuscript, what parts reflect replication and extension of prior work and what contributions are unique to this manuscript. In this context: A key contribution of the manuscript seems to be the connection of an event segmentation pipeline with the scoring of free recall. Could the authors elaborate more on how these two parts connect and potentially position the study's novel contributions based on the relationship between their pipeline and earlier work?

We thank the reviewer for highlighting this important point. We have clarified in the manuscript that this research replicates previous methods but primarily contributes to the connection between event segmentation and automated recall scoring. Specifically, the segmentation components build on Michaelmann et al. (2023) by evaluating newer LLMs, incorporating multiple temperatures, and introducing more detailed agreement analyses. Similarly, recall scoring builds on Heusser et al. (2021) and Raccach et al. (2024) by applying a scalable and easy to implement text embedding approach. As the reviewer noted, a key contribution of our work is the integration of these two components. We highlight that segmentation and memory encoding are not mutually exclusive processes, and thus our approach incorporates these elements into an end-to-end framework. This connection allows us to use LLM-generated event boundaries to anchor memory analyses and provide a direct link to memory performance. We now emphasize these elements in this introduction. Updated excerpts from the introduction are included.

Introduction Page 5: While each of these components – automated segmentation and automated recall – build on prior work, a key contribution of the present study is their integration and expansion.

We use LLM-generated event boundaries to not only assess their alignment with human segmentation, but also as anchors for evaluating free recall, thereby establishing a unified framework that links perception to memory. [...] Page 6: Ultimately, this work extends prior methods to provide an end-to-end framework for automating segmentation and memory recall assessments, enabling scalable analyses of how structured experiences influence memory.

2. The rationale for including temperature as a variable in the segmentation task could be further clarified. While the manuscript explains (in Line 221) that temperature controls output randomness, its conceptual connection to event segmentation was not clear on reading the manuscript. Could the authors explain what the conceptual purpose of increasing randomness in the segmentation is? Specifically, what does the randomness afford besides making models and humans less aligned (which is arguably an undesired outcome with event segmentation and potentially any norming task). The reviewer had addressed a valid concern in the methods section. We have clarified our rationale for making temperature a major focus in our analyses. While temperature controls the randomness of model output, our goal was not only to test consistency in model outputs, but also to explore whether deterministic settings might be too rigid, potentially missing salient boundaries that are obvious to humans.

Methods Page 9: A temperature of 0, being highly deterministic, but may be too rigid and miss some highly salient event boundaries that are evident to humans. By varying the model temperature, we allow the model to generate more variable outputs, helping us test whether it could identify a broad range of meaningful event boundaries while still maintaining overall consistency. This approach ultimately helped us assess both the stability and flexibility of the LLM segmentation behaviours.

3. In Line 274, the paper introduces “shared” boundaries (defined as any overlap between GPT-4 and at least one human) and “distinct” boundaries (no overlap across all instances). Could this definition have led to some of the results where the authors observe higher agreement rates on those boundaries?

We appreciate the reviewer’s thoughtful comment. To clarify, *shared boundaries* are defined as human-marked boundaries that overlap with at least one GPT-4 instance. The proportion of human participants endorsing each of the overlapping boundaries is used as the dependent variable in the analysis. While the *shared boundary* category does require a minimum of at least one human and one model instance matching, our result – that these boundaries have high overall human agreement – is not circular. It indicates that, from the human word-level series, LLMs tend to align with boundaries that are more salient across humans (i.e., more humans identify the boundaries when there is overlap).

4. In Line 324, the authors introduce a normalization based on the minimal and maximal point-biserial correlation. While this measure captures sensitivity to shared boundaries, it was not clear how specificity is afforded by the method. Specifically, a rater (or model) that only places a single event boundary at a point that overlaps with the rater (or model) of comparison would obtain an adjusted correlation of 1, which is the maximum possible correlation that can be obtained from a single response. Could the authors compare their measure with other measures that measure sensitivity and specificity of the segmentation? Relatedly, what is the null distribution of random segmentation?

We thank the reviewer for this comment. To clarify, the method described in Line 324 refers to a transformation of participant confidence rating, which were indicated on a scale from 1 to 10, into a centred scale from -1 to 1 to capture whether they identified a paragraph marking as a true event boundary or a non-boundary and the strength of their decision. This transformation was not a normalization of point-biserial correlations representing whether an event boundary was captured. This method allows us to statistically evaluate participants' *confidence* in judging GPT-identified normative event boundaries compared to non-boundary points (controls). We have updated the manuscript to make this distinction clearer.

Methods Page 13: Subsequently, the ratings were linearly scaled from 1 to 10 to a range of -1 to 1, where -1 represents high confidence that a marking was not an event boundary, 1 represents high confidence that a marking was an event boundary, and 0 represents low confidence for both cases.

5. The current study uses a single fixed prompt throughout the manuscript, so the authors should not claim to test “prompt-level variability” but rather temperature-based variability or sampling consistency.

We would like to clarify that our study does not examine prompt-level variability. The reviewer is correct that a single fixed prompt was used for all API calls, and variability was introduced exclusively through temperature temperature-based sampling. We have ensured that references to variability in the manuscript are framed specifically in terms of temperature-based consistency.

Some minor comments include:

1. In the introduction, LLaMA is presented as a key open-source and privacy-preserving alternative to GPT-4, which helps motivate the study. However, it is then largely dropped from the main analyses—with limited explanation—being excluded from the human validation and recall scoring components. The brief mention that its performance was “notably poorer” may benefit from further elaboration. Given its framing in the introduction, it would be helpful to more clearly justify its exclusion from later stages and to clarify its intended role in the study. If the goal is to generalize findings across LLMs, providing more detail on LLaMA's performance and limitations would strengthen that claim.

We believe this is an excellent suggestion. We have updated the description in the Human Ratings of Normative Boundaries Procedure and Recall to better reflect our justification for focusing solely on GPT-4 generated event boundaries.

Methods Page 12: Although LLaMA 3.0 was initially considered due to its open-source and privacy-preserving advantages, it was not included in these subsequent analyses because its segmentation performance was notably poorer than GPT-4. Given these limitations, we focused on GPT-4 with a temperature of 0, which yielded the most consistent and human-aligned results (described below).

Methods Page 16: In line with the results above, GPT-4 alone was used in these recall assessment methods since its event boundary responses evidently outperformed that of LLaMA 3.0.

2. In Line 563, the manuscript notes that prior work did not account for model variability across prompts; however, the cited study (Michelmann et al.) did include multiple runs.

We thank that reviewer for noting this oversight. We corrected this and explicitly described the differences in the approaches despite the similarities in executing multiple runs.

Discussion Page 22: Previous work has used a prompt-based segmentation approach with GPT 3.0 (Michelmann et al., 2023), which included multiple model runs to assess reliability; however, each run was treated independently, focusing on the alignment individual runs and human annotations, and evaluating the consistency of this effect across several single-run instances. In contrast, our approach evaluates how a group of model runs collectively relate to human responses by treating each model run as an individual instance and allowing us to mirror the structure of our human analyses. Although the previous work relying primarily on responses from a single run of GPT-3.0 with a fixed temperature setting of 0 under the assumption that the model would produce highly reliable and deterministic outputs, it did not explicitly quantify within-model consistency or variability across iterations.

3. Line 140, The manuscript refers to introducing “new analysis metrics,” but it is somewhat unclear which elements are novel contributions versus adaptations of prior methods.

We agree with this reviewer’s comment. To avoid confusion, we have removed the word “novel” from this sentence.

Introduction Page 3: In this paper, we extend and validate methods for investigating event segmentation and subsequently leverage its properties for application in recall assessments.

4. There are some instances where the wording could be more precise. For instance, several findings are framed in terms of “LLMs” more broadly, though the analyses rely only on GPT-4. In another instance, in Line 388, the manuscript refers to “true boundaries,” which might be more accurately described as “GPT-identified boundaries.”

We appreciate the reviewer's comment and agree that greater specificity could be helpful. We have revised several instances where results were previously attributed to LLMs to now more accurately reflect that analyses rely specifically on GPT-4, included the specific section noted by the reviewer.

In Figure 3B, the title appears to reference LLaMA, but the plot title is "Human-to-GPT". This may be a labeling error.

This has been corrected.

5. The y-axes across subplots in Figure 3 use different scales, which makes it harder to visually compare effects across conditions. Standardizing the y-axis range across these plots would improve readability and support more intuitive comparisons.

We have updated the figures to more appropriately represent the comparisons between GPT-4 and LLaMA 3.0.

6. In Figure 3C, I was interested in understanding why the proportions of shared and distinct boundaries do not sum to one. A brief note on how these values are computed mathematically would help readers interpret the plot.

In line with reviewer #2's comments, we have generated a method figure which should help to communicate how the metric is generated. In short, the metric in Figure 3C, is not a measure of the proportion of shared and distinct boundaries, but rather the proportion of participants that have identified boundaries at locations that are shared or unique to LLM-generated boundaries. We hope that the methods figure helps to clear this confusion.

7. In Line 432 and Figure 5, the manuscript mentions that recall and narrative matrices are resized to the median number of segments for visualization. It would be helpful to clarify whether this resizing occurs before or after computing the recall similarity scores.

This is a helpful suggestion. To clarify, the transformation is performed after computing the narrative \times recall matrix. Specifically, the matrices for each model and narrative pair are averaged across participants, and this resulting matrix is resized to 13×13 for visualization. This has been updated in the figure caption.

Figure 7 Page 22: For visualization of the average matrices across narratives, matrices were transformed to square matrices based on the median number of narrative events (13×13). Resizing was performed after computing recall similarity scores for each narrative and does not affect the original analysis.

8. In Line 590, the manuscript notes that "LLaMA 3.1 [models] are associated with substantial hardware costs... too large even for a powerful computer." It would be helpful to specify which model size this refers to. For example, the 8B version of LLaMA 3.1 can typically run on a high-end consumer machine,

and the authors may be referring to the 70B model. Since consumer hardware can differ, it would also be more informative to specify the exact hardware requirements directly.

We have clarified that our comment refers specifically to LLaMA 3.1 70B, which exceeds the hardware limits of our computer system. We have revised the manuscript to include this distinction and specify our hardware specifications.

Discussion Page 25: Although LLaMA evades potential privacy concerns associated with proprietary models because it can be run locally, newer LLaMA models, such as LLaMA 3.1 70B, are associated with substantial hardware costs and memory requirements as it has become too large even for a powerful computer with a high-grade graphics card to manage. Despite using a high-end consumer system (32 GB RAM and an NVIDIA RTX 4080 GPU with 16 GB VRAM), we were unable to run the 70B parameter LLaMA 3.1 locally.

Typos

Line 294: repeated use of “for which.”

This has been corrected.

Line 362: “his” should read “this.”

This has been corrected.

Figure 4 title: “conference rating” should be “confidence rating.”

This has been corrected.

Line 398: “be recommend” should be “be recommended.”

This has been corrected.

REVIEWER #2

The manuscript describes an approach to segmenting human recall of events using LLMs, rather than traditional approaches that rely on human raters. They then validate this model by applying it to human recall data.

First, I should note that this is not my domain of expertise. The methods are interesting, and I broadly support publication, but I cannot comment on how this work fits in with the literature space (either related methodological approaches or theoretical aspects of event-related memory). However, the manuscript is well-written, and I understand the core technical aspects sufficiently. Below, I have listed several points that I think could improve the manuscript.

- In describing the segmentation metrics (p. 8-12), it is easy to get lost, in part because the metrics are described in natural language. It would help the reader to include equations (or perhaps visualizations) to clarify each metric’s calculation and how they are all distinct (some of them sound quite similar at first!). This is in contrast to the results, which are much easier to digest at a glance because the authors have provided many easy-to-understand figures.

The reviewer highlights a strong and important points regarding the clarity of the segmentation metrics. We agree that the natural description along may make it difficult to distinguish between difference computational methods. In response, we have developed a methods figure (Figure 2) that visually demonstrates how the data is manipulated at each step, as well as the outcome measure. We believe these additions will significantly enhance the clarity and accessibility of the methodological details.

A. Agreement Index

Binary word-level series for each participant

$$\text{Agreement Index} = \text{corr}(\text{participant}, \text{average of others})$$

B. Human-to-LLM Agreement

$$\text{Human-LLM Agreement} = \text{corr}(\text{participant}, \text{LLM average})$$

C. Shared vs. Distinct Boundaries

Shared Boundary (Human + LLM) Distinct Boundary (Human only)

$$\text{Amplitude}_{\text{shared}} \text{ vs. } \text{Amplitude}_{\text{distinct}}$$

D. Between-Group Consistency

100 permutations

$$\text{Consistency} = P_{\text{matching}}$$

2. Most analyses make use of mixed-effect models, but the random effect structure could be clarified. First, a glance at the code on github suggests the authors primarily use crossed-intercept models, but do not include random slopes (this isn't clear in the text). Best practice is to use a maximal model justified by the design (Barr et al., 2013) and iteratively reduce this structure when encountering singular fit or convergence issues. Random intercept only models are known to inflate Type I errors (which is probably why they are so common, despite not being best practice). The authors should revise or justify these choices. Second, the authors should justify any discrepancies between models. There are some cases where they use two random intercepts (narrative and participant) and others where they use only one. It might be obvious why to the authors, but in trying to make sense of it for the first time it is a bit confusing. The authors could provide these justifications in each section, or – since many of the analyses parallel each other – include a preliminary paragraph about the mixed effect design and then note where (if) analyses deviate from this.

Barr, D. J., Levy, R., Scheepers, C., & Tily, H. J. (2013). Random effects structure for confirmatory hypothesis testing: Keep it maximal. *Journal of memory and language*, 68(3), 255-278.

We thank the reviewer for this helpful comment. We have updated the text to better reflect our use of random intercept only models throughout. In accordance with best practices (Barr et al., 2013), we initially tested maximal random effects structures justified by the design; however, the random slopes frequently resulted in singular fits or convergence issues. We also note that in some analysis, such as the agreement index, only a single observation per participant per narrative was available, making it impossible to include subject-level random effects. To ensure model stability and comparability across analyses, we adopted a simplified structure using random intercepts only.

In addition, we have added justification for differences in random effects across models. Specifically, for within-subject analyses, random intercepts were included for both subject and narrative to account for variability at each level. For between-subject analyses (e.g., agreement index which measures consistency within group agreement), only narrative-level intercepts were included, as subject was not a repeated measure or analyses were conducted on group-level data.

Methods Page 9: The random effects structure in the linear mixed effects models were tailored to the design of each analysis. For within-subject analyses, we included random intercepts for subject and narrative to account for variability at both levels. For between-subject analyses, random intercepts were included only for narrative. This is noted and justified in the corresponding methods sections. Although a maximal random structure was initially tested where justified by the design, nearly all models resulted in singular fits. As a results, we adopted a simplified structure with random intercepts

only (i.e., removal of random slopes) to ensure model stability and comparability across models (Barr, 2013; Barr et al., 2013).

3. In the introduction, the authors generally talk about “perception” and “spatiotemporal information”. They should make clear early on (and in the abstract) that their methods are related to text or transcribed speech data. I assumed that might be the case but was not too confident until I reached the Methods. There are plenty of experiments in which participants recall narratives after watching videos, but the methods described would ignore any non-text data and probably wouldn’t be suitable. We thank the reviewer for this helpful suggestion. We agree that clarifying the modality of our materials earlier in the manuscript is essential for the reader to correctly interpret the applicability of the methods. We have updated the abstract and introduction to explicitly indicate that our study focuses on narratives presented through text and transcribed spoken recall. These changes help to distinguish our approach from studies that rely on audiovisual materials and clearly signpost our use of text-based data.

Abstract Page 2: To address these concerns, we leverage Large Language Models (LLMs) to automate event segmentation and assess recall of written narratives, employing chat completion and text-embedding models, respectively.

Introduction Page 3: In this paper, we focus specifically on written narratives and spoken recall transcripts, to extend and validate methods for investigating event segmentation and subsequently leverage its properties for application in recall assessments. [...] Page 5: In the current research, we leverage LLMs to automate event segmentation and recall assessments using narrative texts and their corresponding spoken recall.

4. The motivation for the work could be strengthened.
 - a. First, the authors point out that there are several other methods to accomplish the same task using HMMs, topic models, and at least one that uses LLMs. But no comparisons to these models are made. Can these models be applied to the same data or not? Are they worse than the authors’ method? The authors claim they are harder to implement, but that's all.

We appreciate the reviewer’s observation regarding other methods such as topic modelling and HHMs. While these approaches have proven effective in prior work (e.g., Heusser et al., 2021), their computational goals and structures differ meaningfully from our current approach. Topic modelling captures latent thematic content based on word co-occurrences, and when combined with HMMs, can model how these topics shift over time; however, our study is motivated by the theoretical relationship between event segmentation and memory encoding – extensive research suggests that segmentation supports how experiences are

recalled in the future. Accordingly, our method evaluates recall by aligning participant free recall with discrete event boundaries. This allows us to assess memory in terms of structured event-based content, rather than abstract topic distributions. While topic modelling may be applicable in some contexts, it does not directly address the foundations of segmentation in recall. Similarly, the LLM-based methods used in Martinez (2024) applied GPT-4 to score clausal recall of short passages using highly constrained outputs with predetermined targets. This approach differs from ours in both structure and aim and is less suited for evaluating the free recall of longer narratives and the assessment of overall gist comprehension. We have updated manuscript to better exemplify our motivation (i.e., the relationship between event segmentation and recall) and inability to directly compare previous methods.

Introduction Page 5: While each of these components – automated segmentation and automated recall – build on prior work, a key contribution of the present study is their integration. We use LLM-generated event boundaries to not only assess their alignment with human segmentation, but also as anchors for evaluating free recall, thereby establishing a unified framework that links perception to memory. [...] Page 6: Ultimately, this work extends prior methods to provide an end-to-end framework for automating segmentation and memory recall assessments, enabling scalable analyses of how structured experiences influence memory.

Discussion Page 26: While some of these recent techniques have achieved near-perfect correlations to manual raters ($r \approx 0.99$), they have primarily used short narrative passages (~60 words, Chandler et al., 2021; Martinez, 2024), where scores based on constrained outputs and a predetermined set of details rather than overall gist (Chandler et al., 2021; Martinez, 2024; Raccah et al., 2024; Van Genugten & Schacter, 2024). [...] Methods such as topic modelling and HMM have also been successfully applied (Heusser et al., 2021); however, their aims differ from the current approach. Specifically, such methods focus on modeling latent thematic transitions over time, whereas our study is explicitly grounded in the theoretical relationship between event segmentation and recall.

- b. Second, the authors frequently refer to this as an “automated” approach (including in the title) but I don’t really see how. Sure, I can copy the prompt to do event segmentation – easy enough. But for recall assessment, I need to programmatically use a language model to encode responses, create a correlation matrix, etc. Essentially, I need to reproduce the author’s analysis pipeline. To their credit, they have their code on github, and it is fairly clear, but hosting code doesn’t make it automated. When I hear “automated” what I expect is a tool where

researchers can plug in a .csv of their data (and possibly an OpenAI token) and it spits out results that can be analyzed in the standard fashion. My experience in authoring an open-source methods tool in my own area has taught me that if you actually want people to *use* your method, you have to make it dead easy – including for those who study event memory but know little about LLMs and are not methodologists. Otherwise, people will mostly only cite your paper in the way that the authors cite previous HHM and topic model approaches – something that has been done in the past, not a method that is being currently used. For example, the authors could cobble together a bunch of functions from their existing code into a single .R file (and from there, it is easy enough to host it as a library on CRAN!) I think this would be a boon for the authors as well, who would surely get more citations if their methods are actually used.

(Relatedly, the authors state that “manual scoring guidelines and approaches may differ between research groups possibly leading to inconsistencies in the literature and reproducibility challenges.” Proposing yet *another* method does not seem like it really solves that problem.)

This is perhaps the most time-consuming critique of mine to address, but I think it is also the most central for a methods paper where the goal certainly feels like convincing your readers to use this method. I encourage the authors to strongly consider it. If the goal is only to see whether LLMs segment data like humans, the paper needs to be reframed - but I find that framing much less appealing.

The reviewer brings a fair perspective regarding the *automated* nature of the method we propose in this paper. We agree that while code availability is essential, automation implies a streamlined and easy-to-implement interface, especially for researchers who may not have experience working with large language models.

To address this, we have taken several steps. We have developed a module that allows users to input a text path for their narrative and recall data and automatically returns segmentation and recall scores in a standard data frame with easy export to csv capabilities. This includes functions for both isolated segmentation and segmentation-based recall scoring. The GitHub repository now contains a detailed README file with step-by-step instructions and syntax examples.

We hope that these updates better align with the reviewer’s expectations for an automated pipeline and demonstrate our commitment to usability and reproducibility. We believe that this will improve methodological adoption, addressing the reviewer’s comments regarding long-term impact.

Please reference <https://github.com/ryanapanela/EventRecall>.

5. The analyses at temperature 0 are confusing. I typically assume that a temperature parameter of 0 in softmax means deterministic. They even use this word when referring to Michelmann et al.'s approach on p. 22. But from the figures, the results are clearly not deterministic. In Fig 2A, there is clearly variability in GPT-4. In Fig 2C, LLaMA is *almost* deterministic, though there appears to be one event boundary that not all instances agreed on. Relatedly, I would expect the within-group agreement for temperature 0 to be perfect (in fact, I questioned why the authors needed to run it 20 times). But the results in Fig 3 reveal that it is not. It would help to clarify why this is the case.

We thank the reviewer for mentioning this. While temperature 0 is typically described as producing deterministic outputs, LLMs can still exhibit minor variability in their responses due to factors such as token sampling, caching, and other nondeterministic processes at the API-level. To clarify this point, we have updated the methods and discussion to explicitly note that although temperature 0 is expected to produce stable results, perfect reducibility cannot be guaranteed; hence, our motivation for executing the 20 runs. We also emphasize that from our assessment, variability observed was minor and does not compromise the reliability of the model outputs. These clarifications aim to address the reviewer's concerns while reinforcing that low-temperature settings still offer strong alignment to human responses.

***Methods Page 14:** A temperature of 0, while generally considered deterministic, may still exhibit variability due to stochastic factors within the API – such as nondeterministic behaviours in token sampling or caching (Atil et al., 2025). At the same time, a temperature of 0, may be too rigid and miss salient event boundaries that are evident to humans. By varying the model temperature, we allow the model to generate more variable outputs, helping us test whether it could identify a broad range of meaningful event boundaries while still maintaining overall consistency. This approach ultimately helped us assess both the stability and flexibility of the LLM segmentation behaviours.*

***Discussion Page 24:** Lower temperatures resulted in more deterministic and human-aligned segmentation outputs; however, we did observe some variability across instances, even in the lowest temperature conditions. This aligns with recent work showing that LLMs can exhibit nondeterministic behaviours due to token sampling and API-level processing (Atil et al., 2025), indicating that although variability is minor, exactly replicability cannot be assumed. Obtaining normative event boundaries from the average responses across LLM instances, may be advantageous.*

6. (p. 19+) The split-half reliabilities are fairly low, but the LLM/human correlations are noticeably lower. Is this an issue? In simple linear regression, standardized betas are the same as Pearson correlation (the authors include random effects here so I'm not so sure how that affects the interpretation). Here,

they are reported in the range of [.36, .52]. So, approximately, R^2 in the range of [.13, .27]. Fine – but not amazing. I will leave it to other reviewers who know more about the field, but in absence of any comparisons to other methods cited, it's hard to tell whether I should be impressed by this. Perhaps the authors could comment on it.

We appreciate this thoughtful comment and have clarified our interpretation. Specifically, we emphasize that our standardized β coefficients are derived from linear mixed effects models, which unlike a standard linear regression, are not equivalent to Pearson correlations, which were commonly reported in prior work. Accordingly, these values are inherently more conservative than correlations, which make them comparable to correlation-based estimates from similar open-ended recall tasks (e.g., Van Genugten & Schacter, 2024). We have also expanded the discussion to better contextualize our results with the literature. Notably, although some studies have reported near-perfect correlations (e.g., Martinez, 2024), making our results seem subpar, we highlight that many of these studies have used constrained outputs, short narrative passages and recall phrases, or pre-defined narrative details. These methodological differences limit their direct comparability to our approach, which involves longer free recall and evaluates gist of narrative events. These changes are now reflected in the discussion.

Page 26: While some of these recent techniques have achieved near-perfect correlations to manual raters ($r \approx 0.99$), they have primarily used short narrative passages (~60 words, Chandler et al., 2021; Martinez, 2024), where scores based on constrained outputs and a predetermined set of details rather than overall gist (Chandler et al., 2021; Martinez, 2024; Racciah et al., 2024; Van Genugten & Schacter, 2024). [...] Our observed standardized coefficients ($\beta = 0.37 - 0.52$), derived from linear-mixed effects models are inherently more conservative than Pearson correlations typically reported in prior work, and thus are appropriately comparable to estimates from other work using similar open-ended recall tasks (e.g., $r \approx 0.60$, Van Genugten & Schacter, 2024).

Minor:

1. p. 8 the authors state that higher temperature values are “more creative”. This is a strong take, I would remove.

This has been corrected.

2. p. 8 Authors should clarify what “max_tokens” is in plain language

We agree that explaining the *max_tokens* parameter will provide more clarity to the methods. This has been updated in the methods.

Page 9: We set the max_tokens parameter to 4096 to avoid incomplete responses. This parameter defines the maximum number of tokens – units of text, such as words or punctuations – that the model can generate. A high allocation ensures that the model can produce a segmentation response for the entire narrative.

3. Fig 3B Should “Human-to-GPT” be “Human-to-LLaMA” ?

This has been corrected.

4. Fig 7B I suggest adding either an identity line or at least rescaling the axes to be the same. Correlations are important, but absolute agreement is important too, and it’s harder to discern from these figures with different axes.

We thank the reviewer for highlighting this oversight. As the reviewer mentioned, correlations are important. We have updated the figures to include regular x- and y-axes to better visualize the relationship between automated and human scores.

REVIEWER #3

My task was to review the code and ensure that it runs and is documented well.

1. README

There isn't enough information in the README.md file. The only information is:

"This repository contains data and files used for: Panela, R. A., Barnett, A. J., Barensen, M. D., & Herrmann, B. (2025). Event Segmentation Applications in Large Language Model Enabled Automated Recall Assessments (Version 1). arXiv. <https://doi.org/10.48550/ARXIV.2502.13349>"

A README.md file should explain what's in the repository. It should explain that there are both R and Python files. It should state what output they both produce, and it should explain how to install the dependencies and run them.

For R using RStudio automatically prompts and installs packages. This is not always the case for Python. It is expected that a requirements.txt file or some other means of tool/library management is outlined. To produce a requirements.txt files, the following can be run on the command line if a virtual environment was used and the libraries/packages were installed using pip.

```
pip freeze > requirements.txt
```

See below for the output under "# Python code" .

It would be good to outline in the README what data is included and what data isn't -- I could produce types of graphs, but not all the graphs in the paper. This could possibly be because not all the data was included. If all the data was indeed included, it wasn't clear to me how to produce all the outputs and not the types of outputs.

We thank the reviewer for this comment. The README file on GitHub has been updated to more accurately reflect the contents of the repository. Along with the comments from Reviewer #2, the repository also includes a useable module that can be imported into Python.

2. R code

When the data files are used in the code it would be good to have some signposting such as "PATH_TO/" to remind the user to insert the absolute path to the files or a variable could be defined so the user would only have to change the path to the data file in one place.

A directory signpost has been added to the data import portion of the scripts.

In the data file "gpt_segmentation_data.csv", the column "word_number" is referred to in the file "gpt_segmenation_analysis.Rmd" on line 64, after the data is ingested. However, there is no "word_number" column, only a "word_numbers" column in the .csv data file. Perhaps this could be corrected.

We apologize for this oversight. The code has been corrected.

In the data file "visual_time_word_data.csv", the column "temperature" is referred to in the file "gpt_segmenation_analysis.Rmd" on line 157, after the data has been ingested. However, there is no "temperature" column in that .csv file at all. Therefore, the code between lines 153 to 302 could not be run at all. If somehow the column "temperature" had to be adjoined to the data file, this was not clear.

This has been corrected and should run without issue.

In gpt_segmentation_analysis.Rmd, the code after line 302 could be run, but because the variable "temperature" (see above) is used after this point. However, the "Identified event boundaries" graphs (Figure 2 Segmentations in the paper) do not look like the graphs that I could reproduce (because previous segments of the code could not be run as mentioned above).

With the above correction, the remainder of the script functions without issue. We apologize for the oversight and thank the reviewer for pointing this out. We have also verified that these issues are not present in the llm_segmentation_analysis.Rmd script.

The R code files are not as well commented as the Python code is. There are headings for each of the sections, but no explanation of the code as there are in the Python files.

Additional comments have been added in the RMarkdown file.

3. Python Code

In the README file, it would be good to know if the Python code is expected to be run first, or the R code or whether it doesn't matter.

We thank the reviewer for this comment. The Rmd script are provided mainly to reproduce the statistics and plots provided in the manuscript. The Python scripts, especially for the segmentation section, are provided as reference to carry out the method; however, it is worth noting that due to model variation, executing the gpt_llama_segmentation.py script will not produce segmentation results that exactly match our results. The preprocessed data is provided in the data folder on GitHub.

Additionally, it would be good to state which Python file to run. One file (segmentation_functions.py) serves as resources/tools file, while the file the "gpt_llama_segmentation.py" file is the main file to be executed.

There is an error in the definition for `get_finish_reason`. It is defined on line 164 as the following with 2 arguments:

```
get_finish_reason(responses, choice):...
```

However, when it is called on line 237, it has 1 argument.

```
get_finish_reason(item)
```

Therefore, the Python code threw the following error:

```
TypeError: get_finish_reason() missing 1 required positional argument: 'choice'
```

Once I had fixed this in the code, I reran it.

The reviewer has rightfully identified the lack of a default parameter. This has been corrected with the default parameter set to 0, as per API syntax.

As mentioned above, it would be good what dependencies are required for the Python code with their required version because this is one way to ensure the longevity of the code. My output when I run "pip freeze > requirements.txt" after installing all the dependencies was:

```
distro==1.9.0
exceptiongroup==1.2.2
filelock==3.18.0
fonttools==4.57.0
fsspec==2025.3.2
h11==0.14.0
httpcore==1.0.8
httpx==0.28.1
huggingface-hub==0.30.2
idna==3.10
Jinja2==3.1.6
jiter==0.9.0
kiwisolver==1.4.8
MarkupSafe==3.0.2
matplotlib==3.10.1
mpmath==1.3.0
networkx==3.4.2
numpy==2.2.5
openai==1.76.0
packaging==25.0
pandas==2.2.3
pillow==11.2.1
psutil==7.0.0
pydantic==2.11.3
pydantic_core==2.33.1
pyparsing==3.2.3
python-dateutil==2.9.0.post0
pytz==2025.2
PyYAML==6.0.2
regex==2024.11.6
requests==2.32.3
safetensors==0.5.3
scipy==1.15.2
seaborn==0.13.2
six==1.17.0
sniffio==1.3.1
sympy==1.13.3
tokenizers==0.21.1
torch==2.7.0
tqdm==4.67.1
transformers==4.51.3
typing-inspection==0.4.0
typing_extensions==4.13.2
tzdata==2025.2
urllib3==2.4.0
```

After installing the required libraries and changing the file paths, I was still not able to run the Python code because of the following error:

Traceback (most recent call last):

```
File "/Users/misticam/Projects/event_recall/code/segmentation/produce_segmentation/gpt_llama_segmentation.py", line 10, in <module>
  llama_events = llama_segmentation(f'/Users/misticam/Projects/event-recall/data/stories/{story}.txt', iters=iterations, temperature=temp)
```

```
File "/Users/misticam/Projects/event-recall/code/segmentation/produce_segmentation/segmentation_functions.py", line 284, in
  llama_segmentation
  curr_response = prompt_llama(curr_prompt, temperature=temperature)
```

```

File "/Users/misticam/Projects/event-recall/code/segmentation/produce_segmentation/segmentation_functions.py", line 134, in
prompt_llama
    response = pipe(prompt_message, temperature=temperature)

File "/Users/misticam/Projects/event-recall/envs/review/lib/python3.10/site-packages/transformers/pipelines/text_generation.py", line 280,
in __call__
    return super().__call__(Chat(text_inputs), **kwargs)

File "/Users/misticam/Projects/event-recall/envs/review/lib/python3.10/site-packages/transformers/pipelines/base.py", line 1379, in
__call__
    return self.run_single(inputs, preprocess_params, forward_params, postprocess_params)

File "/Users/misticam/Projects/event-recall/envs/review/lib/python3.10/site-packages/transformers/pipelines/base.py", line 1386, in
run_single
    model_outputs = self.forward(model_inputs, **forward_params)

File "/Users/misticam/Projects/event-recall/envs/review/lib/python3.10/site-packages/transformers/pipelines/base.py", line 1286, in
forward
    model_outputs = self._forward(model_inputs, **forward_params)

File "/Users/misticam/Projects/event-recall/envs/review/lib/python3.10/site-packages/transformers/pipelines/text_generation.py", line 385,
in _forward
    output = self.model.generate(input_ids=input_ids, attention_mask=attention_mask, **generate_kwargs)

File "/Users/misticam/Projects/event-recall/envs/review/lib/python3.10/site-packages/torch/utils/_contextlib.py", line 116, in
decorate_context
    return func(*args, **kwargs)

File "/Users/misticam/Projects/event-recall/envs/review/lib/python3.10/site-packages/transformers/generation/utils.py", line 2358, in
generate
    prepared_logits_processor = self._get_logits_processor()

File "/Users/misticam/Projects/event-recall/envs/review/lib/python3.10/site-packages/transformers/generation/utils.py", line 1194, in
_get_logits_processor
    processors.append(TemperatureLogitsWarper(generation_config.temperature))

File "/Users/misticam/Projects/event-recall/envs/review/lib/python3.10/site-packages/transformers/generation/logits_process.py", line 282,
in __init__
    raise ValueError(except_msg)

ValueError: `temperature` (=0) has to be a strictly positive float, otherwise your next token scores will be invalid.

```

In the paper "temperature=0" is one of the hyperparameters that is shown, so there are some discrepancies. The above ellipses "..." in the pasted error simply are my local file paths. I am happy to rerun the code when it gets updated. I think this repo would be a great resource for other researchers once the issues are resolved.

We again thank that reviewer for this comment. GPT and LLaMA are functionally different in how they manage temperature parameters. GPT allows for true zero temperature, using greedy decoding to select the single most likely next token. As the reviewer mentioned, LLaMA only allows positive values, as it selects the next token by sampling from a probability distribution. For the LLaMA implementation we used a temperature of 0.1 which is effectively the lowest permissible and functions in a nearly deterministic manner. This value is functionally equivalent to a temperature 0. We have also elaborated on our motivation for varying the temperature parameters, explaining that while lower temperature promote consistency, they may be too ridged and overlook some of the salient event

boundaries. The `segmentation_functions.py` code has been updated to ensure that only positive temperature parameters are present for LLaMA.

***Methods Page 8:** Although OpenAI's API allows a true zero temperature, generating text based on the highest probability (i.e., greedy decoding), the LLaMA implementation requires a strictly positive temperature, as it generates text by sampling from a probability distribution. We used 0.1, the lowest permissible value, which is effectively deterministic. This value is functionally equivalent to a temperature 0, and we refer to it as such throughout for consistency with GPT-4. It is worth noting that a temperature of 0, while generally considered deterministic, may still exhibit variability due to stochastic factors within the API – such as nondeterministic behaviours in token sampling or caching (Atil et al., 2025). Nonetheless, temperature 0 (or 0.1 for LLaMA) provides functionally deterministic outputs and is suitable for assessing model consistency. At the same time, a temperature of 0, may be too rigid and miss salient event boundaries that are evident to humans.*

REVIEWER #3

My part of the review was to ensure that the code runs smoothly. The follow up task was to ensure that the Python code ran. The installation instructions don't work for me as expected.

I ran the command from the README:

```
pip install git+https://github.com/ryanapanela/EventRecall.git
```

I created a script to run with the first code snippet under the heading "Python API Usage" and got the following error:

```
ModuleNotFoundError: No module named 'segmentation'
```

I then used tried to install via "setup.py" by running:

```
python setup.py build
python setup.py install
```

And still got the same error.

We thank the reviewer for testing our module and indicating its failures. We have updated the setup.py script to ensure that the segmentation and recall modules are findable during installation.

I installed the dependencies in requirements.txt and ran the code snippet again. When I got it running I got the error:

```
FileNotFoundError: [Errno 2] No such file or directory: 'test/Run.txt'
```

The file Run.txt is in the folder data/stories/. Once I moved the file to a folder named "test" in the same folder as my script, the could get the code snippet from the README running.

This has been corrected to ensure the code references the location of Run.txt in the repository.

The automatic installation packaging doesn't work, but the core code runs once you add your own API key.

It would be good to know which version of Python is required in the README.

This has been corrected.

Requirements: Python 3.8+